# Modularity, criticality, and evolvability of a developmental gene regulatory network

**Berta Verd[1,2,3,4†]\*, Nicholas AM Monk[5], Johannes Jaeger[1,2,3,5,6,7,8,9‡]\***

[1]EMBL/CRG Systems Biology Research Unit, Centre for Genomic Regulation (CRG), The Barcelona Institute of Science and Technology, Barcelona, Spain; [2]Universitat Pompeu Fabra (UPF), Barcelona, Spain; [3]Konrad Lorenz Institute for Evolution and Cognition Research (KLI), Klosterneuburg, Austria; [4]Department of Genetics, University of Cambridge, Cambridge, United Kingdom; [5]School of Mathematics and Statistics, University of Sheffield, Sheffield, United States; [6]Wissenschaftskolleg zu Berlin, Berlin, Germany; [7]Center for Systems Biology Dresden (CSBD), Dresden, Germany; [8]Complexity Science Hub (CSH), Vienna, Austria; [9]Centre de Recherches Interdisciplinaires (CRI), Paris, France

**\*For correspondence:**
bertaverd@gmail.com (BV);
yoginho@gmail.com (JJ)

**Present address:** [†]Department of Genetics, University of Cambridge, Cambridge, United Kingdom; [‡]Department of Molecular Evolution and Development, University of Vienna, Vienna, Austria

**Competing interests:** The authors declare that no competing interests exist.

**Abstract** The existence of discrete phenotypic traits suggests that the complex regulatory processes which produce them are functionally modular. These processes are usually represented by networks. Only modular networks can be partitioned into intelligible subcircuits able to evolve relatively independently. Traditionally, functional modularity is approximated by detection of modularity in network structure. However, the correlation between structure and function is loose. Many regulatory networks exhibit modular behaviour without structural modularity. Here we partition an experimentally tractable regulatory network—the gap gene system of dipteran insects—using an alternative approach. We show that this system, although not structurally modular, is composed of dynamical modules driving different aspects of whole-network behaviour. All these subcircuits share the same regulatory structure, but differ in components and sensitivity to regulatory interactions. Some subcircuits are in a state of criticality, while others are not, which explains the observed differential evolvability of the various expression features in the system.
DOI: https://doi.org/10.7554/eLife.42832.001

## Introduction

Systems biology aims to understand the function and evolution of complex regulatory networks. This requires some sort of hierarchical decomposition of these networks into manageable and intelligible subsystems, whose properties and behaviour can be analysed and understood in relative isolation (*Simon, 1962*; *Riedl, 1975*; *Lewontin, 1978*; *Bonner, 1988*; *Raff, 1996*; *West-Eberhard, 2003*; *Schlosser and Wagner, 2004*; *Callebaut et al., 2005*). If each subsystem possesses a clearly delimited and discernible function, the network can be subdivided into *functional modules* (*Raff, 1996*; *von Dassow and Munro, 1999*; *Hartwell et al., 1999*; *Wagner et al., 2007*; *Mireles and Conrad, 2018*). In the Introduction of our paper, we provide a careful argument showing that the most common approach to identify functional modules has severe limitations, and propose an alternative method, which we then use to dissect and analyse a specific pattern-forming network, the gap gene system of the vinegar fly, *Drosophila melanogaster*.

The most common strategy to identify functional modules is to partition the graph representing a network into simple motifs (*Shen-Orr et al., 2002*; *Alon, 2007*) or subcircuits (also called subnetworks or communities; (*Girvan and Newman, 2002*; *Oliveri and Davidson, 2004*; *Babu et al.,*

*2004*; *Levine and Davidson, 2005*; *Newman, 2006*; *Davidson and Erwin, 2006*; *Oliveri and Davidson, 2007*; *Erwin and Davidson, 2009*; *Davidson, 2010*). Network motifs are small subgraphs that are identified through their statistical enrichment (*Alon, 2007*; *Alon, 2006*), while subcircuits are characterised by a high connection density among their component nodes contrasting with sparse connections to the outside (*Girvan and Newman, 2002*; *Radicchi et al., 2004*; *Newman, 2006*; *Wagner et al., 2007*; *Fortunato, 2010*). In both cases, subsystems are defined in terms of the regulatory structure or network topology: they are *structural modules*. This approach presupposes a strong connection between functional and structural modularity (see, for example, *Lim et al., 2013*).

Strictly interpreted, structural modules are mutually exclusive: they are disjoint subgraphs of a complex regulatory network that do not share nodes between each other (*Girvan and Newman, 2002*; *Radicchi et al., 2004*; *Palla et al., 2005*). And yet, such modules can never be fully isolated: their context within the larger network influences behaviour and function. The structural approach therefore relies on the assumption that context-dependence is weak, and structural modularity is generally pronounced enough, to preserve the salient properties and behaviour of a motif or subcircuit in its native network context.

Structural modularity is widely regarded as a necessary condition for the evolvability of complex networks. 'Evolvability,' in the general sense of the term, is defined as the ability to evolve (*Dawkins, 1989*; *Wagner and Altenberg, 1996*; *Hendrikse et al., 2007*; *Pigliucci, 2008*). More specifically, evolvability refers to the capacity of an evolving system to generate or facilitate adaptive change (*Wagner and Altenberg, 1996*; *Pigliucci, 2008*). Structural modularity can boost this capacity in several ways. Entire modules can be co-opted into new pathways during evolution, generating innovative change (*Raff, 1996*; *von Dassow and Munro, 1999*; *True and Carroll, 2002*; *Davidson and Erwin, 2006*; *Erwin and Davidson, 2009*; *Monteiro and Podlaha, 2009*; *Wagner, 2011*). Furthermore, each module can vary relatively independently, and it has been argued that this accounts for the individuality, origin, and homology of morphological characters as well as their trait-specific variational properties (*Wagner and Altenberg, 1996*; *Wagner et al., 2007*; *Wagner, 2014*). Finally, structural modularity allows for a fine-tuned response to specific selective pressures by minimizing off-target pleiotropic effects (*Wagner and Altenberg, 1996*; *Pavlicev et al., 2008*; *Wagner and Zhang, 2011*).

The identification and analysis of structural modules has been very successful in many cases. For example, it has been used to understand the regulatory principles of segment determination in *Drosophila* (*von Dassow et al., 2000*; *Ingolia, 2004*), the origin and evolution of butterfly wing spots (*Carroll et al., 1994*; *Brakefield et al., 1996*; *Keys et al., 1999*; *Beldade et al., 2002*; *Monteiro et al., 2003*; *Monteiro et al., 2006*) and beetle horns (*Moczek, 2006*), and the mechanism and evolution of larval skeleton formation in sea urchins and sea stars (*Hinman et al., 2003*; *Hinman and Davidson, 2007*; *Oliveri et al., 2008*; *Gao and Davidson, 2008*). Other examples abound in the literature (see *Raff, 1996*; *Schlosser and Wagner, 2004*; *Callebaut et al., 2005*; *Peter and Davidson, 2015* for comprehensive reviews).

In spite of its usefulness, structural modularity has a number of serious limitations. Some modelling studies suggest that it is not necessary for evolvability (see, for example, *Crombach and Hogeweg, 2008*). Furthermore, it is notoriously difficult to identify structural modules and delimit their boundaries with any precision. One reason for this may be that the definition of (sub)system boundaries is fundamentally context- and problem-dependent (see, for example, *Chu et al., 2003*; *Chu, 2011*). More to the point, even the simplest subcircuits tend to exhibit a rich dynamic repertoire comprising a range of different behaviours depending on context (boundary conditions), quantitative strength of parameter values (determining genetic interactions as well as production and decay rates), and the specific form of the regulation-expression functions used to integrate multiple regulatory inputs (*Mangan and Alon, 2003*; *Wall et al., 2005*; *Ingram et al., 2006*; *Siegal et al., 2007*; *Payne and Wagner, 2015*; *Ahnert and Fink, 2016*; *Perez-Carrasco et al., 2018*; *Page and Perez-Carrasco, 2018*). Because of this, it is usually not possible to single out subgraphs exhibiting specific behaviours and functions that are robustly independent of their native network context. In cases like these, looking for structural modules is not the most fruitful approach to subdivide a complex regulatory network.

A recent simulation-based screen of multifunctional gene regulatory networks nicely illustrates this important point (*Jiménez et al., 2017*). The authors performed a systematic computational

search for network structures able to implement two qualitatively different dynamical behaviours. They then identified the particular subcircuits that were responsible for either of the two behaviours (or functions). What they found is an entire spectrum of structural overlap among functional modules. At one end of the spectrum, 'hybrid' networks represent the sum of two completely disjoint structural modules. At the other end, 'emergent' networks use exactly the same nodes and connections to implement both dynamical behaviours (note that *Salazar-Ciudad et al., 2000*; *Salazar-Ciudad et al., 2001a*; *Salazar-Ciudad et al., 2001b*) introduce a different definition of 'emergent' network structure, indicating a flat hierarchy of regulatory interactions rich in regulatory feedback). Most networks identified by the screen fell somewhere in between these two extremes, that is, they show partial structural overlap between functional modules. This suggests that most functionally modular networks are not modular in the strict structural sense.

The limitations of structural modularity can be further illustrated using the real-world example of the gap gene system. This gene regulatory network is involved in pattern formation and segment determination during the blastoderm stage of early embryogenesis in dipteran insects such as *D. melanogaster* (see *Jaeger, 2011* for review). Its regulatory structure is summarized in *Figure 1A*. Nodes in this network represent genes encoding transcription factors that activate or repress each other as indicated by their connections. The gap gene network reads and interprets morphogen gradients formed along the major or antero-posterior (A–P) axis of the embryo by the protein products of the maternal co-ordinate genes *bicoid (bcd)*, *caudal (cad)*, and *hunchback (hb)*. This results in broad, overlapping expression domains for the trunk gap genes *hb*, *Krüppel (Kr)*, knirps (kni), and *giant (gt)* (*Figure 1B, C*). Extensive gap-gene cross-regulation, especially during the last cleavage cycle (cycle 14A, C14A) of the blastoderm stage, is essential for the correct dynamic positioning of expression domain boundaries, in particular the dynamic kinematic shifts of posterior gap domains towards the anterior of the embryo (*Jaeger et al., 2004b*; *Jaeger et al., 2004a*; *Surkova et al., 2008*; *Manu et al., 2009a*; *Crombach et al., 2012*; *Verd et al., 2018*). Maternal co-ordinate and gap genes together then regulate pair-rule and segment-polarity genes, which form a molecular pre-pattern at the time of gastrulation that precedes and determines the formation of morphological segments during later stages of development.

Due to its small size and high connection density (see *Figure 1A*), it is not possible to identify structural clusters in the gap gene system. Previous simulation-based analyses identified a number of mechanisms driving gap gene expression. One of these consists of two double-negative (positive) feedback loops between *hb/kni* and *Kr/gt* that are responsible for the basic staggered arrangement of gap domains along the A–P axis. Another one comprises the asymmetric repressive interactions between overlapping gap genes (*e. g. kni* on *Kr* and *Kr* on *hb*) that are driving dynamic anterior shifts of gap domains over time (*Figure 1A, B*) (*Jaeger et al., 2004a*; *Jaeger et al., 2004b*; *Jaeger, 2011*; *Crombach et al., 2012*). Note that these mechanisms are not structurally modular, as they involve different interactions among the same set of genes. Moreover, attempts to explain gap gene expression dynamics with network motifs have consistently failed to correctly account for the integrated behaviour of the whole system (*Zinzen and Papatsenko, 2007*; *Ishihara and Shibata, 2008*; *Papatsenko, 2009*).

All these difficulties arise because the gap gene network is a strongly emergent network (*sensu Jiménez et al., 2017*). If there are functional modules in this network, they show a high degree of structural overlap. For such emergent networks, analyses based on structural modularity fail, because this approach is only valid for systems of markedly hybrid character. As was argued in *Jiménez et al. (2017)*, such systems only constitute a tiny subset of all possible naturally occurring regulatory networks.

An alternative way to identify network modules that does not rely directly on their structure is to define them in terms of their activity (*Kauffman, 1993*; *Hartwell et al., 1999*; *von Dassow and Munro, 1999*; *Newman and Bhat, 2008*; *Newman and Bhat, 2009*; *Alexander et al., 2009*; *Benítez and Alvarez-Buylla, 2010*). Such *dynamical modules* or dynamical subsystems consist of a group of connected network nodes that implement a particular behaviour (*Irons and Monk, 2007*; *Benítez and Alvarez-Buylla, 2010*). Their spatio-temporal pattern of activity specifies a certain type of dynamics—such as bistable or oscillatory—termed a 'dynamical regime' (*Verd et al., 2017*; *Verd et al., 2018*). Different dynamical regimes are distinguished by the composition of the underlying phase portraits (*Strogatz, 2015*; *Perez-Carrasco et al., 2018*). For instance, they can be generated by different subsets of a system's attractors (with associated basins) (*Irons and Monk, 2007*).

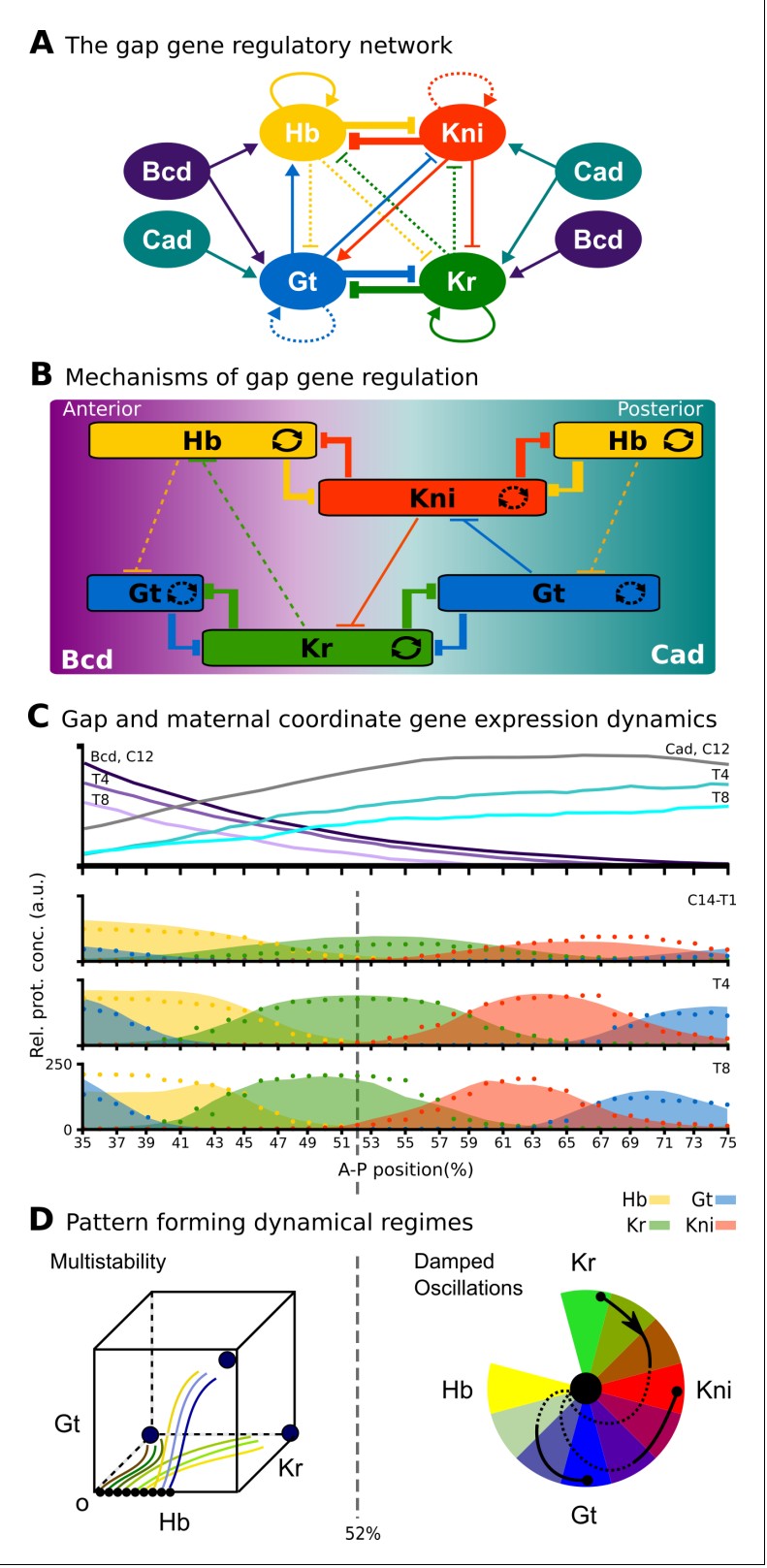

**Figure 1.** The gap gene system: structure, regulation, and expression. (**A**) The regulatory structure of the gap gene network. Nodes represent transcription factors encoded by maternal coordinate genes *bicoid* (*bcd*, purple) and *caudal* (*cad*, cyan), as well as trunk gap genes *hunchback* (*hb*, yellow), *Krüppel* (*Kr*, green), *knirps* (*kni*, red) and *giant* (*gt*, blue). Activating interactions are indicated by arrows, repressive interactions by T-bars. Circular arrows

*Figure 1 continued on next page*

*Figure 1 continued*

indicate auto-activation. Dashed lines represent weak, solid lines stronger regulatory interactions. (**B**) Spatial representation of gap gene regulation: boxes indicate relative positions of gap domains along the antero-posterior (A–P) axis of the embryo. Anterior is to left, posterior to the right. Background colour indicates predominant activating inputs by maternal gradients. T-bars indicate gap gene cross-repression. Circular arrows represent gap gene auto-activation. (**C**) Gene expression dynamics of maternal coordinate genes *bcd* (purple) and *cad* (cyan), as well as trunk gap genes *hb* (yellow), *Kr* (green), *kni* (red) and *gt* (blue). Quantified expression patterns are shown as lines (maternal coordinate) and coloured areas (gap genes). Output of the fitted full model is shown as dots. Y-axes represent relative protein concentration in arbitrary units (a.u.); X-axes represent A–P position in %, where 0% is the anterior pole. C12 and C14 refer to cleavage cycle 12 and 14A, respectively. C14A is subdivided into eight time classes T1–8 of equal length. Only the trunk region of the embryo from 35 to 75% is shown. (**D**) Dynamical regimes driven by the gap gene system. Multistable (switch-like) behaviour in the anterior is indicated by a phase space with trajectories converging to multiple point attractors (represented by circles). Damped oscillations leading to dynamic anterior shifts of gap domains in the posterior are indicated by a colour wheel with trajectories cycling through successions of different gap genes. The bifurcation boundary between the two regimes at 52% A–P position is indicated by a dashed line. See text for details. Figure adapted from *Verd et al. (2017)*; *Verd et al. (2018)*.

DOI: https://doi.org/10.7554/eLife.42832.002

With this type of classification, nodes may be shared between overlapping modules, simultaneously driving different dynamics in the context of different subsystems. No disjoint modular network structure is required or even expected (*Jiménez et al., 2017*). Still, each module's behaviour is relatively independent and clearly distinguishable from that of other dynamical subsystems (*Irons and Monk, 2007*).

Note that dynamical modules are not the same as *co-expression modules* (sometimes also called regulatory modules; see, for example, *Eisen et al., 1998*; *Bar-Joseph et al., 2003*; *Segal et al., 2003*; *Stuart et al., 2003*; *Kim et al., 2009*; *Koch et al., 2017*). The latter are defined by the correlated or anti-correlated expression of their components. In contrast, the mutual dependence of component expression patterns in dynamical modules is causal, rather than correlative. It can be very complex and obscured by the non-linearity of regulatory interactions. Dynamical modules are not identified by individual expression patterns, but rather by the coherent collective activity of the module as a whole (*Kauffman, 1993*; *Irons and Monk, 2007*; *Alexander et al., 2009*).

Just like their structural counterparts, dynamical modules also influence evolvability. However, the way they achieve this is fundamentally different. Structural subcircuits can vary independently since they are only loosely interconnected. Dynamical modules, in contrast, drive distinct behaviours that exhibit different levels of sensitivity to changes in system parameters (*Alexander et al., 2009*). In particular, some dynamical modules may be in a state of *criticality*, close to a bifurcation point beyond which their dynamics may change drastically and abruptly; others may be structurally stable, that is, not critical and far from any bifurcation, and therefore insensitive or robust to changes in parameters (*Thom, 1976*; *Kauffman, 1993*; *Jaeger et al., 2012*; *Jaeger and Monk, 2014*; *Jaeger and Sharpe, 2014*; *Strogatz, 2015*). Recall that specific network components may be involved in governing more than one dynamic behaviour. Mutations affecting these components will tend to have a strong effect on those behaviours that are critical, leaving others unaltered because they are structurally stable. This type of dynamic modularity implies that the network is more likely to evolve in certain directions than others. While mutations may be random, their effects on network dynamics are certainly not. Behaviours driven by modules that are critical (close to bifurcation) will be more labile than those driven by modules that are structurally stable. This amounts to functional modularity, even though the overall structure of the network is not modular.

Dynamical modularity transcends the inherent limitations of structural approaches and allows us to gain insights into heavily emergent regulatory networks. However, it is far from trivial to apply this framework to the experimental study of specific evolving developmental processes. Irons and Monk have developed an algorithmic method to identify dynamical modules in Boolean network models (*Irons and Monk, 2007*). Unfortunately, this method is difficult to generalize and adapt to continuous mathematical frameworks (for instance, models formulated as systems of ordinary differential equations) used to study morphogen-driven pattern-forming networks such as the gap gene system.

For this reason, we adopt a pragmatic approach to identify dynamical modules in the gap gene network of *D. melanogaster*. Our approach is based on the observation that only subsets of gap genes are expressed and exert their regulatory influence in any one region of the embryo. This allowed us to identify four localized subsystems, each containing three trunk gap genes that are active in different but overlapping regions of the embryo. Surprisingly, all four subsystems share the same regulatory structure, which identifies them as AC/DC subcircuits (*Panovska-Griffiths et al., 2013*; *Perez-Carrasco et al., 2018*). AC/DC subcircuits represent one of the simplest known genetic systems able to produce both switch-like (multistable) and oscillatory behaviour. We show that these AC/DC modules drive distinct dynamical regimes: static domain boundaries in the anterior, versus anteriorly shifting gap domains in the posterior of the embryo (*Figure 1C, D*; *Verd et al., 2017*; *Verd et al., 2018*). The boundary between these two dynamical regimes is positioned by an AC/DC circuit in a state of criticality. This makes the position of this boundary especially sensitive to changes in the strength of regulatory interactions, and we argue that this has shaped the evolvability of the gap gene system within the Dipteran order (flies, midges, and mosquitoes).

## Results and discussion

### Modularity

#### The gap gene system is composed of three dynamical modules

We have taken a pragmatic approach to identifying dynamical modules in the gap gene network (*Figure 1A, B*; *Jaeger, 2011*). The analysis asks which of the four trunk gap genes (*hb*, *Kr*, *kni*, and *gt*) are required—or, more accurately, which ones are *not*—to drive correct expression dynamics in nuclei at different positions along the antero-posterior (A–P) axis during cleavage cycle 14A (C14A). It is based on a detailed dynamical model of the gap gene network (the 'full model'), which has been fit to quantitative spatio-temporal data of trunk gap gene expression (see *Figure 1C*) (*Verd et al., 2017*; *Verd et al., 2018*). This model implements accurate dynamic mechanisms of gap gene regulation that have been extensively validated against empirical evidence (*Jaeger et al., 2004b*; *Jaeger et al., 2004a*; *Manu et al., 2009b*; *Manu et al., 2009a*; *Ashyraliyev et al., 2009*; *Jaeger, 2011*; *Crombach et al., 2012*; *Verd et al., 2017*; *Verd et al., 2018*). We consider the sensitivity of the network to a gap gene negligible, if the node representing that gene in the network can be removed from the model at the onset of C14A without significant consequences to the resulting expression dynamics (see 'Node sensitivity analysis' in 'Materials and methods'). If sensitivity is negligible in a given nucleus, we conclude that the gene is not required to drive gap gene expression in that nucleus during C14A.

The results of our analysis are shown in *Figure 2A*. They reveal three regions that are insensitive to specific gap genes: (1 in the region between 35 and 47% A–P position, developmental trajectories are insensitive to *kni* (red background); (2) between 49 and 59% A–P position, they are insensitive to *gt* (blue), and (3) between 61 and 75% A–P position, they are insensitive to *hb* (yellow). Therefore, the gap gene regulatory network can be reduced from four to three trunk gap genes in each of these three regions (*Figure 2A and B*). Regional boundaries reflect the position of expression boundaries, but differ from those in that they remain constant, while expression patterns change over time during C14A (*Surkova et al., 2008*; *Jaeger, 2011*).

Next, we single out a minimum set of gap gene interactions that are still able to drive correct expression dynamics in each of these three regions. Surprisingly, the structure of all three resulting subcircuits is qualitatively the same, even though each involves a different (overlapping set of gap genes (*Figure 2B*), and therefore distinct strengths of regulatory interactions (*Figure 2C*). This structure combines a negative feedback loop between all three genes of a subcircuit with a double-negative (positive) feedback loop between two of them (*Figure 2B, C*). Such a combination between positive and negative feedback loops is called the 'AC/DC circuit,' first described in the context of dorso-ventral patterning in the vertebrate neural tube (*Balaskas et al., 2012*; *Panovska-Griffiths et al., 2013*; *Perez-Carrasco et al., 2018*). Positive feedback loops occur between non-overlapping gap genes Kr and Gt, as well as Hb and Kni; the interactions involved are much stronger than the negative repressive interactions between gap genes with overlapping expression domains (*Figure 2C*). Previous work has shown that strong positive feedback is required for the basic staggered arrangement of gap domains ('alternating cushions'), while weaker (and hence slower)

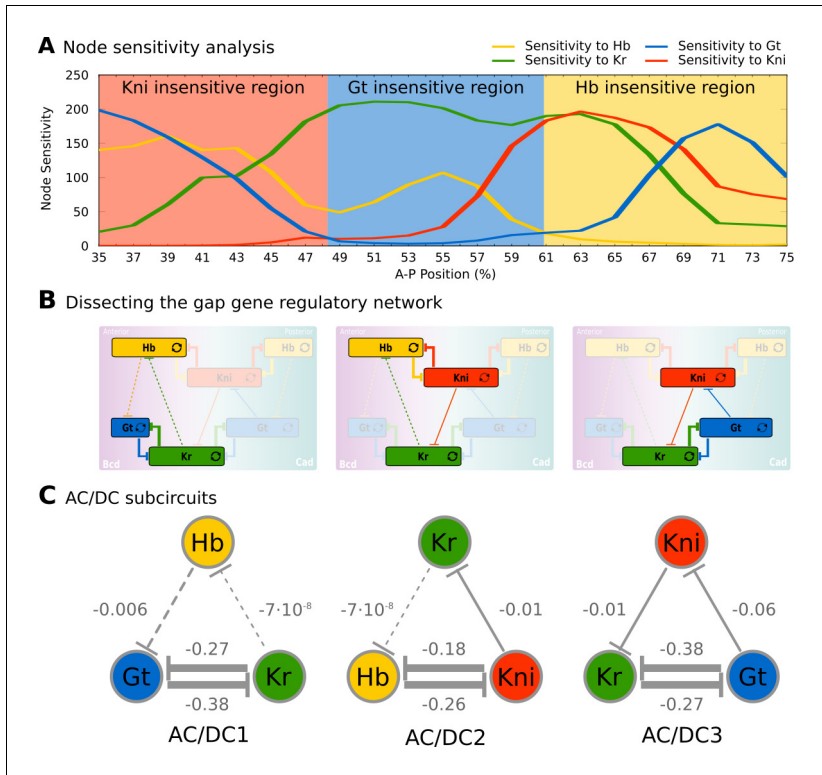

**Figure 2.** Identification of dynamical modules in the gap gene system. (**A**) Node sensitivity analysis. The plot shows sensitivity of model output to the removal of *hb* (yellow), *Kr* (green), *kni* (red) and *gt* (blue). Y-axis represents node sensitivity as defined in 'Materials and methods,' *Equation (4)*; X-axis represents A–P position in %, where 0% is the anterior pole. Regions insensitive to the absence of specific gap genes are indicated by background colour. (**B**) Dissecting the gap gene network into dynamical modules. Network schemata as in *Figure 1B*. Subcircuits active in each region identified in (**A**) are highlighted. (**C**) AC/DC subcircuits. All subcircuits identified in (**A**) and (**B**) share the same regulatory structure, indicated by T-bar connectors. Numbers indicate strength of interactions (in arbitrary units). Maternal inputs and auto-activation are omitted for clarify. Note that there is a fourth AC/DC subcircuit posterior of the region included in the present analysis (not shown, see 'Materials and methods'). See text for details.

DOI: https://doi.org/10.7554/eLife.42832.003

negative feedback drives anterior shifts in gap domain position over time in the posterior trunk region of the embryo (*Jaeger et al., 2004b*; *Jaeger et al., 2004a*; *Perkins et al., 2006*; *Ashyraliyev et al., 2009*; *Jaeger, 2011*; *Crombach et al., 2012*; *Verd et al., 2018*).

## The dynamics of AC/DC subcircuits faithfully reproduce the dynamics from the full model

The next step is to establish whether AC/DC subcircuits are sufficient for patterning in each of the three embryonic regions identified in *Figure 2A*. In order to qualify as a true dynamical module *sensu Irons and Monk (2007)*, each AC/DC subcircuit must recover the expression dynamics as well as the underlying dynamical regime of the full model in the region where it is active. Anterior to the bifurcation boundary at 52% A–P position, this means static gap domain borders governed by multi-stability; posterior to 52% A–P position, this means kinematically shifting domain boundaries governed by a monostable dynamical regime that drives a stereotypical temporal succession of gap gene expression in each nucleus (*Figure 1B, C* and *Figure 3A*) (*Verd et al., 2017*; *Verd et al., 2018*).

All three AC/DC subcircuits—AC/DC1 in the anterior, AC/DC2 in the middle, and AC/DC3 in the posterior—reproduce the expression dynamics of the full model with reasonable accuracy in their respective regions of influence throughout cleavage cycle 14A, which we subdivide into eight time

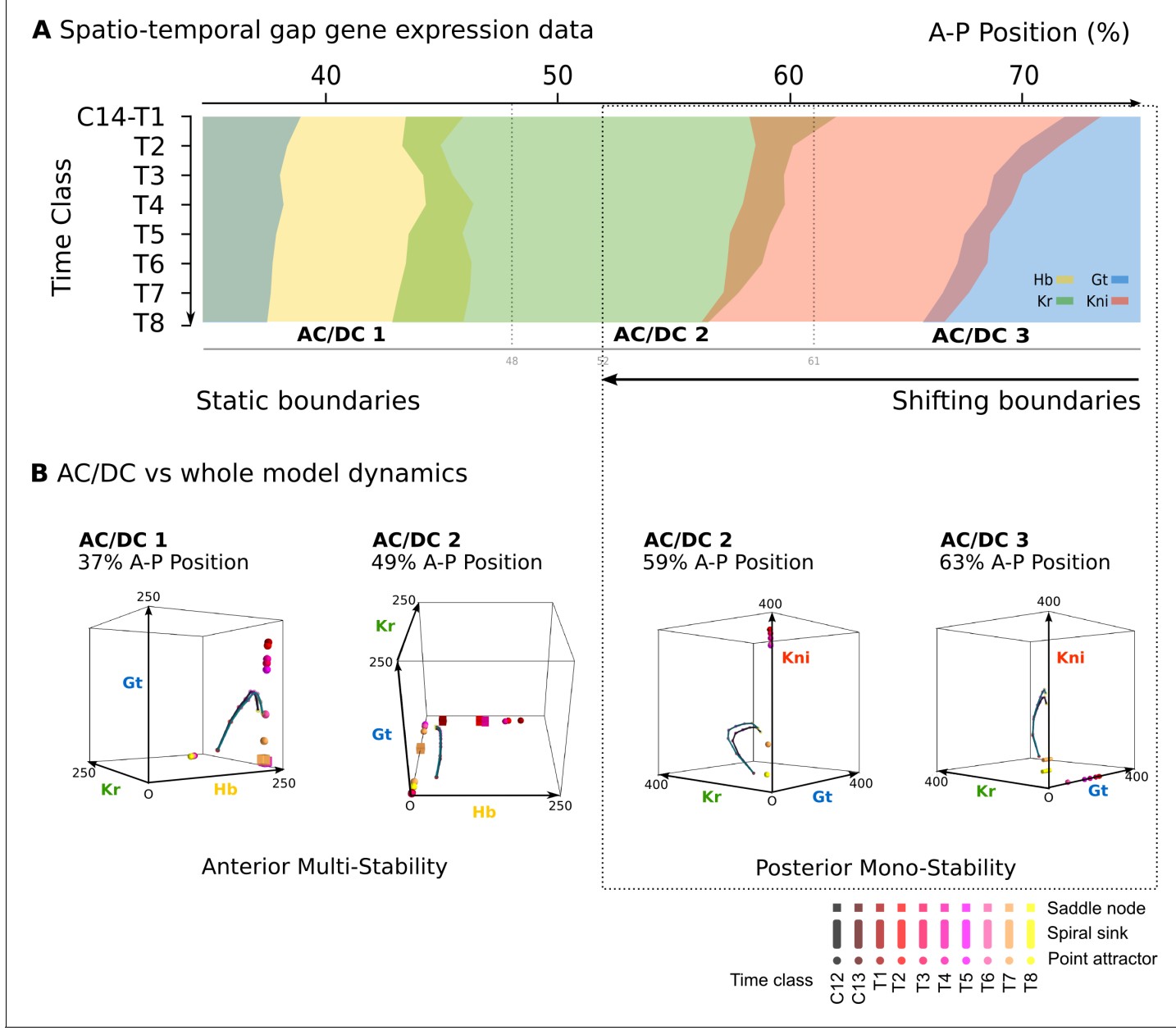

**Figure 3.** AC/DC subcircuits faithfully reproduce the dynamics of the full model. (**A**) Spatio-temporal dynamics of gap gene expression in the trunk region of the embryo. Coloured areas show the position of gap domains for *hb* (yellow), *Kr* (green), *kni* (red), and *gt* (blue). Y-axis represents time (flowing downwards) during cleavage cycle C14A. Time classes T1–eight as defined in **Figure 1**. X-axis represents % A–P position, where 0% is the anterior pole. Regions of influence for each AC/DC subcircuit is indicated by grey lines. Black dotted line indicates the bifucation at 52% position separating static from shifting gap domain boundaries. (**B**) Comparative dynamical analysis of AC/DC subcircuits and the full model. Instantaneous phase portraits of AC/DC1 (nucleus at 37%), AC/DC2 (nuclei at 49 and 59%), and AC/DC3 (nucleus at 63% A–P position) are shown. Point attractors are shown as spheres, spiral sinks as cylinders, saddles as cubes. Colour code indicates time class, from T1 (dark red) to T8 (yellow). Trajectories from simulations of AC/DC subcircuits are shown in turquoise, trajectories from simulations of the full model in black. Axes represent concentrations of gap proteins (in arbitrary units) as indicated. See 'Materials and methods' for model definition and details on phase space analysis.

DOI: https://doi.org/10.7554/eLife.42832.004

The following figure supplements are available for figure 3:

**Figure supplement 1.** Comparison of AC/DC subcircuits and the full model.

DOI: https://doi.org/10.7554/eLife.42832.005

**Figure supplement 2.** Comparative dynamical analysis of AC/DC subcircuits and the full model.

DOI: https://doi.org/10.7554/eLife.42832.006

classes T1–8 of equal length for the purpose of our analysis (*Figure 3—figure supplement 1*) (*Jaeger et al., 2004b*; *Jaeger et al., 2004a*; *Surkova et al., 2008*). The only major expression defect involves the anterior boundary of the posterior *gt* domain, which fails to shift in the AC/DC3 model, starting to deviate from the full model around time class T3 (*Figure 3—figure supplement 1*). Other defects affect aspects of domain shape or levels of expression (*Figure 3—figure supplement 1*), but not the dynamic positioning of boundary interfaces, on which our analysis is focused.

The range of dynamical regimes that can be implemented by a circuit is constrained by its regulatory structure (see 'Introduction'. Which particular dynamical regime is realized depends on the strength of regulatory interactions as well as initial and boundary conditions provided by maternal factors. Since all three AC/DC subcircuits exhibit different interaction strengths (*Figure 2C*) and receive different maternal inputs, they can drive different expression dynamics (see Appendix 1 for a systematic mathematical analysis).

We determine the dynamical regime of each AC/DC subcircuit by calculating and analysing their associated phase portraits. To achieve this, we simulate each AC/DC subcircuit in isolation, including realistic time-variable inputs from maternal gradients (*Verd et al., 2017*) and auto-activation. It has been shown previously that auto-activation does not significantly contribute to patterning in the gap gene system, but is required to regulate accurate levels of gap gene expression (*Perkins et al., 2006*). The time-dependent boundary conditions render these systems time-variant or non-autonomous, and cause their phase portraits to change their geometry over time (*Verd et al., 2017*). Time-variance or non-autonomy means that not only the variables, but also the parameters or boundary conditions change over time, and therefore implicitly the rules governing the regulatory system themselves. Previously, we have developed a successful method for the classification of transient trajectories in time-variant, non-autonomous dynamical systems (*Verd et al., 2014*), and have used it to study the dynamics of gap gene expression in the full model (*Verd et al., 2017*; *Verd et al., 2018*). Here, we redeploy this technique to study each AC/DC subcircuit independently (see 'Materials and methods' for details).

AC/DC1 consists of nodes representing *hb*, *Kr*, and *gt* (*Figure 2C*, left panel. Its region of influence lies between 35 and 47% A–P position (*Figure 2A*). Phase portraits in this region exhibit multistability with two or more point attractors (shown for the nucleus at 37% A–P position in *Figure 3B*, left; additional nuclei are shown in *Figure 3—figure supplement 2A*). These trajectories are very similar to those simulated with the full model, suggesting that flow direction and magnitude is conserved between both models (*Figure 3B* and *Figure 3—figure supplements 2A, A'*). In each case, trajectories are shaped by the pursuit of a moving point attractor (*Verd et al., 2014*), located at equivalent positions in phase space in AC/DC1 and the full model. The only difference between the two models is an additional attracting steady state in the full system, situated at high Kr levels. This attractor is positioned too far from the trajectories to influence their shape.

The nodes in AC/DC2 represent *hb*, *Kr*, and *kni* (*Figure 2C*, central panel), and it is active between 49 and 59% A–P position (*Figure 2A*). This region straddles the bifurcation that occurs at 52% A–P position in the full model, designating the division between static anterior and shifting posterior gap domains (*Manu et al., 2009a*; *Gursky et al., 2011*; *Verd et al., 2017*). To be considered a dynamical module throughout this region, the AC/DC2 subcircuit must recover the two distinct dynamical regimes on either side of the bifurcation boundary. The phase portraits of a nucleus anterior and another one posterior to the bifurcation boundary are shown in *Figure 3B* (centre). Additional nuclei are shown in *Figure 3—figure supplement 2B*.

The nucleus at 49% is located just anterior to the bifurcation. The phase portrait of AC/DC2 in this nucleus is bistable for all time points except the last one (at T8, when it becomes monostable and only an attractor close to the origin persists (*Figure 3B*). This is extremely similar to what happens in the full model, where by T8 the steady states at high Kr values have disappeared, and only steady states close to the origin remain. This transition to monostability occurs too late to significantly alter the course of the trajectory. Instead, the geometry of the trajectory in this nucleus— almost identical in AC/DC2 and the full model—first converges towards a saddle during early C14A, and later directly towards a moving attractor (*Figure 3B*). In the full model, a geometric capture occurs at stages prior to C14A (*Verd et al., 2014*; *Verd et al., 2017*), outside the time range of the AC/DC models (*Verd et al., 2017*).

Steady states in the phase portrait of AC/DC2 at 49% A–P position are restricted to the Hb-Kr plane. They can be mapped onto a subset of steady states present in the full model (*Figure 3—*

*figure supplements 2B, B'*). Additional steady states in the full model are located on the Hb-Gt plane, at positions which are very similar to those present in AC/DC1 in more anterior nuclei (*Figure 3—figure supplement 2A*). This suggests that phase portraits of the full model are composites of AC/DC1 and AC/DC2 in the region anterior to the bifurcation boundary at 52% A–P position.

Posterior to the bifurcation at 52% A–P position, the full model exhibits trajectories that curve towards a spiral sink attractor in a monostable phase portrait (*Verd et al., 2018*). The trajectories of AC/DC2 in this region are very similar (*Figure 3B* for the nucleus at 59% A–P position, see *Figure 3—figure supplements 2B, B'* for trajectories in other nuclei). Both exhibit a spiralling geometry which is confined to the Kr-Kni plane, with some minor deviations in peak concentration at late stages.

Interestingly, however, the underlying topology of phase space is not equivalent in the AC/DC2 subcircuit and the full model. While both models exhibit monostability in this region, and trajectories in both are shaped by the pursuit of a moving attractor, there are no spiral sinks in AC/DC2 (*Figure 3B*, and *Figure 3—figure supplement 2B*). Spiralling trajectories in this subcircuit are shaped mainly by the movement of a conventional point attractor, with spiral sinks appearing only late, at time points T7 and T8. Attractor movement is much more pronounced than in the full model (*Figure 3B*, and *Figure 3—figure supplements 2B, B'*). Furthermore, the steady states of AC/DC2 are located along the Kni axis with early attractors at high concentration values of Kni, while the full model shows steady states arranged along the Gt axis with early attractors at high concentration values of Gt (*Figure 3—figure supplement 2B'*). This is similar to steady states in AC/DC3 in the nucleus at 63% A–P positions (*Figure 3B*, right panel). At later stages, when the influence of attractor position on the geometry of the trajectory becomes more pronounced, the positions of steady states converge between the two models: in each case, they are located at low Kni values, along the Kni axis. This illustrates that identical transient behaviour can be caused by different types of moving attractors in (biological) time-variant or non-autonomous dynamical systems.

The nodes in AC/DC3 represent *Kr*, *kni*, and *gt* (*Figure 2C*, right panel). It is active between 61 and 75% A–P position (*Figure 2A*). In the nuclei at 61 and 63% A–P position, AC/DC3 faithfully reproduces the dynamics of gap gene expression obtained from the full model (*Figure 3B*, right, and *Figure 3—figure supplement 2C*). As in the case of AC/DC2, the system is monostable, and trajectories are shaped by the pursuit of a moving attractor. This attractor is located at high Gt concentrations early on, moving closer to the origin over time with only residual levels of Kni left at late stages (*Figure 3B*, and *Figure 3—figure supplement 2C*). Similar to AC/DC2, the type of attractor differs between the full model, where it is a spiral sink at all time points, and AC/DC2, where it is a conventional point attractor during time classes T1–6, only turning into a spiral sink at time points T7 and T8 (*Figure 3B*, and *Figure 3—figure supplements 2C, C'*).

Posterior to 63% A–P position, AC/DC3 no longer recapitulates gap gene expression dynamics accurately, because its trajectories fail to switch from the Kr-Kni to the Kni-Gt plane as they do in the full model (*Figure 3—figure supplement 1C*). It is possible that this switch requires overlap with a fourth AC/DC subcircuit in the posterior sub-terminal region of the embryo, which could not be characterised further since most of its region of influence lies outside the spatial range we can analyse in the full model (see 'Materials and methods').

In summary, phase space analysis of AC/DC subcircuits establishes that they are true dynamical modules of the gap gene network in the region between 35% and 63% A–P position (*Irons and Monk, 2007*). They faithfully recover the geometry of trajectories in the full model, whose phase portrait can be seen as an overlapping composite of those of the subcircuits in this region. This simple picture is complicated by the fact that spiralling trajectories occur posterior to 52% A–P position despite the absence of spiral sinks in the phase portraits of AC/DC models. The lack of spiral sinks is compensated by more pronounced movements of conventional point attractors, suggesting that distinct time-variant or non-autonomous phase space topologies can generate equivalent transient dynamics. We discuss this somewhat degenerate relationship between phase space topology and trajectory shapes further in the 'Conclusions.'

## Criticality and evolvability

AC/DC1 in the anterior and AC/DC3 in the posterior each have a consistent dynamical regime across their respective regions of influence (*Figures 1–3*), indicating that these two dynamical modules are structurally stable with respect to inputs from maternal gradients. In contrast, AC/DC2 correctly reproduces the bifurcation observed in the full model at 52% A–P position, which separates static

anterior patterning from kinematically shifting domains in the posterior (*Figure 3*) (*Manu et al., 2009a*; *Gursky et al., 2011*; *Verd et al., 2017*). The presence of a bifurcation implies that this dynamical module is in a state of criticality with respect to the inputs from maternal gradients. In other words, the parameters of the AC/DC2 circuit place it very close to a bifurcation boundary, which it will cross due to the different maternal inputs it receives along the A–P axis of the embryo within its region of influence. Our analysis therefore suggests that the gap gene network of *D. melanogaster* is critical to maternal inputs only in the middle of the embryo, where AC/DC2 is active between 49 and 59% A–P position, while it is structurally stable outside of this region. This contrasts with an earlier proposition—based on the quantification of cross-correlations between expression patterns and a set of purely theoretical models of gap gene regulation—which suggested that the system shows signs of criticality along the entire antero-posterior axis (*Krotov et al., 2014*).

Understanding criticality in complex regulatory networks is far from trivial, since bifurcations may depend on more than just one of the system's parameters (see, for example, *Thom, 1976*; *Scheffer, 2009*; *Kuznetsov, 2004*). This can lead to counter-intuitive effects. For instance, gap gene patterning in the central region of the *D. melanogaster* blastoderm is robust towards variation in the levels of maternal gradients (*Figure 4*) (*Houchmandzadeh et al., 2002*; *Gregor et al., 2007a*; *Manu et al., 2009b*; *Gursky et al., 2011*). This is surprising in light of the fact that AC/DC2 is in a critical state. In contrast, the position of the bifurcation boundary between static and shifting gap domains is labile between different dipteran species (*Jaeger et al., 2004b*; *Surkova et al., 2008*; *Manu et al., 2009a*; *García-Solache et al., 2010*; *Jaeger, 2011*; *Crombach et al., 2014*; *Wotton et al., 2015*). In the scuttle fly *Megaselia abdita* (Phoridae), it is located more anteriorly compared to *D. melanogaster*: the region where gap domain shifts occur includes the Hb-Kr interface at around 40% A–P position (see *Figure 5* below) (*Wotton et al., 2015*). The moth midge *Clogmia albipunctata* (Psychodidae) shows even more extended and pronounced gap domain shifts (*García-Solache et al., 2010*; *Crombach et al., 2014*). In the following subsections, we will provide an analysis that resolves this apparent paradox.

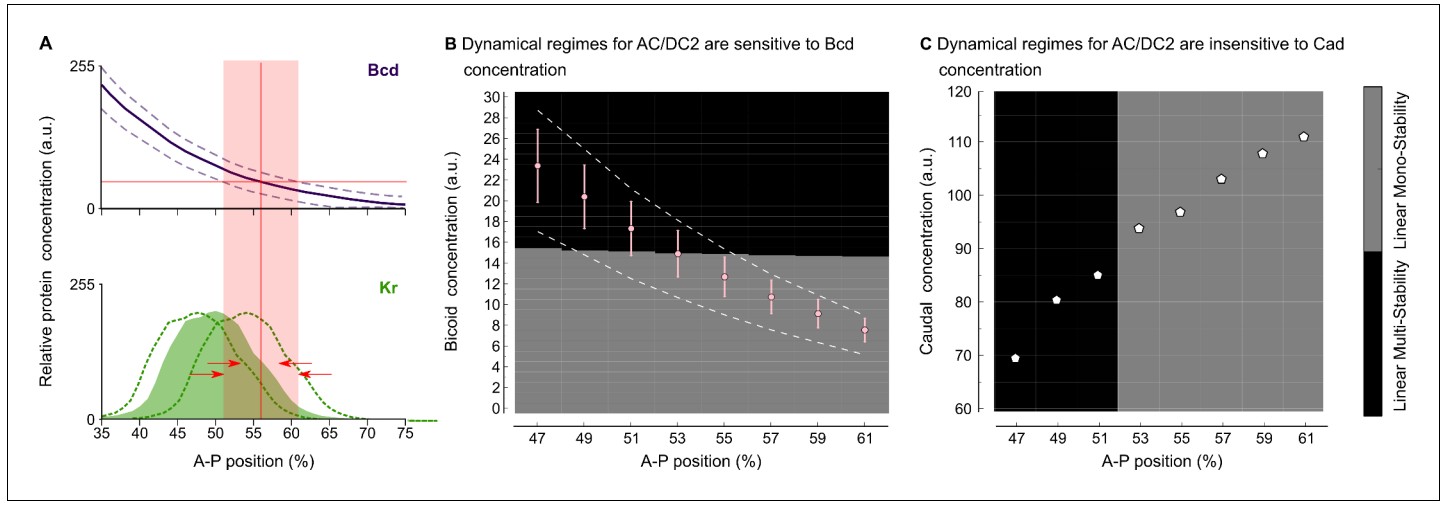

**Figure 4.** Intra-species robustness of gap gene patterning to perturbations in levels of maternal gradients. (**A**) The position of gap gene expression features is relatively robust towards changes in maternal gradient concentrations. This sketch—modified from Figure 1 of *Manu et al. (2009b)*—shows concentration variation in the maternal Bcd gradient (above) and the zygotic expression domain of Kr (below). Solid line and coloured area indicate averaged expression patterns, dashed lines indicate expression variation. Red arrows show the difference between the range of positional variation in Bcd (red background) and Kr. (**B**) Phase diagram for AC/DC2 in response to variation in Bcd concentration. Pink dots and error bars show average Bcd concentration with standard deviation between 47 and 61% A–P position. Dashed lines show the maximum range of Bcd profiles in the data. AC/DC2 subcircuits for nuclei indicated by the X-axis, where simulated with Cad concentration fixed to its value at T1, and Bcd concentration fixed to the values given by the Y-axis. Background colour indicates the resulting dynamical regime: the multistable anterior regime is shown in black, the monostable posterior regime in grey. (**C**) Phase diagram for AC/DC2 in response to variation in Cad concentration. White dots indicate Cad concentrations between 47 and 61% A–P position. AC/DC2 subcircuits for nuclei indicated by the X-axis, where simulated with Bcd concentration fixed to its value at T1, and Cad concentration fixed to the values given by the Y-axis. Background indicates dynamical regime as in (**B**). See text for details.
DOI: https://doi.org/10.7554/eLife.42832.007

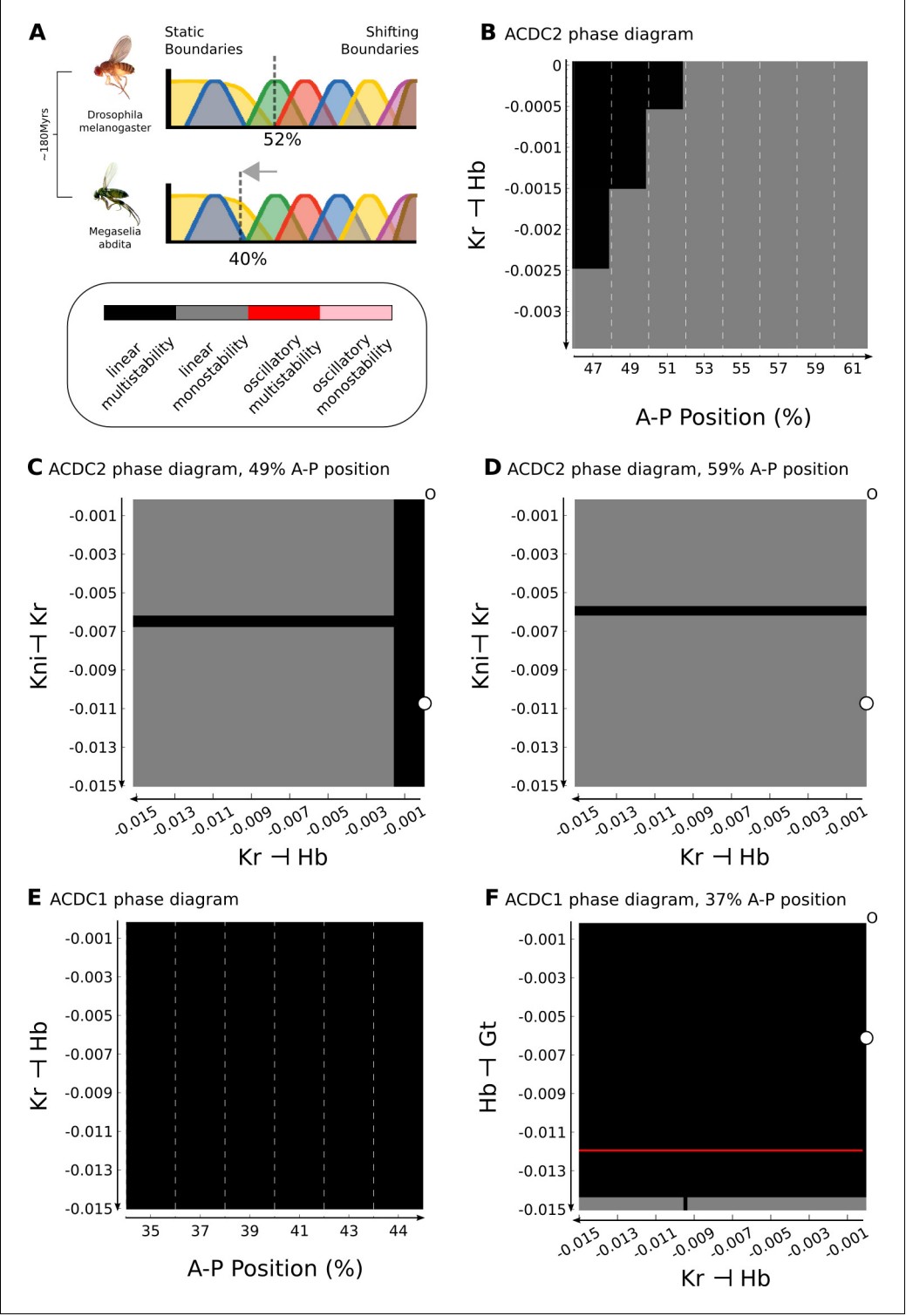

**Figure 5.** Inter-species lability of the bifurcation boundary depends on altered gap-gap interactions. (**A**) The position of the bifurcation boundary separating static and shifting gap gene expression domains differs between *D. melanogaster* (upper panel, 52%) and *M. abdita* (lower panel, 40% A–P position). The difference between species is highlighted by a grey arrow. Phylogenetic distance between the two species is indicated to the left in 'million years ago' (mya). (**B**) Phase diagram for AC/DC2 in response to altering the strength of *hb* repression by Kr (plotted against A–P position). (**C, D**) Phase diagram for AC/DC2 in response to altering both *hb* repression by Kr (X-axis) and *Kr* repression by Kni (Y-axis) shown for subcircuits in nuclei at 49 (**C**) and 59% (**D**) A–P position. (**E**) *Figure 5 continued on next page*

*Figure 5 continued*

Phase diagram for AC/DC1 in response to altering the strength of *hb* repression by Kr (plotted against A–P position). (F) Phase diagram for AC/DC1 in response to altering both *hb* repression by Kr (X-axis) and *gt* repression by Hb (Y-axis) shown for the subcircuit in the nucleus at 37% A–P position. (A–F) Background indicates dynamical regimes as in *Figure 4*: the multistable anterior regime is shown in black, the monostable posterior regime in grey. In addition, there is a narrow strip of a multistable oscillatory regime in (F) (shown in red). All Phase diagrams were calculated with maternal gradient concentrations fixed to their values at T1. See text for details.

DOI: https://doi.org/10.7554/eLife.42832.008

## Intra-specific robustness to maternal gradient concentration

We used the AC/DC2 model to assess the effect of changing the maternal Bcd and Cad gradients on the position of the bifurcation boundary (cf. *Figure 4*). In the region between 47 and 61% A–P position, Bcd spans a range of concentration from 24 to 7 arbitrary units in our data (*Figure 4B*). For each nucleus in this region, we fixed Cad levels to their values during early C14A (time point T1), while varying Bcd levels in steps of 0.01 from to 30 arbitrary units. We then calculated the phase portraits of AC/DC2 for each combination of maternal input concentrations (see 'Materials and methods').

The resulting dynamical regimes are shown in the phase diagram displayed in *Figure 4B*. There is a threshold response at a Bcd concentration of approximately 15 arbitrary units: above this threshold, the system is bistable; below, it is monostable. Bcd concentrations in nuclei anterior of 52% fall into the multistable regime, nuclei posterior of this position fall into the monostable regime (see dots in *Figure 4B*), recovering the same bifurcation position as in the full model (*Manu et al., 2009a*; *Gursky et al., 2011*; *Verd et al., 2017*).

Quantitative measurements of the Bcd gradient reveal a standard deviation of about 15% relative concentration (*Gregor et al., 2007b*; *Liu et al., 2013*). Plotting this as error bars in *Figure 4B*, we find that they cross the boundary between dynamical regimes only in nuclei between 51 and 53% A–P position. This very narrow spatial range is in agreement with observed levels of variability in the data and in previous analyses of gap gene circuits (*Gregor et al., 2007b*; *Manu et al., 2009b*; *Gursky et al., 2011*). If we alter the concentration of Bcd by a larger amount, as happens in mutants with varying numbers of *bcd* copies (*Driever and Nüsslein-Volhard, 1988*; *Houchmandzadeh et al., 2002*; *Liu et al., 2013*), we see a more pronounced displacement of the boundary as indicated by the intersection of the dashed gradient profiles with the bifurcation threshold in *Figure 4B*.

The situation is very different for the maternal Cad gradient. In the region between 47 and 61% A–P position, Cad spans a range of concentrations from 70 to 110 arbitrary units (*Figure 4C*). For each nucleus in this region, we fixed Bcd levels to their values during early C14A (time point T1), while varying Cad levels in steps of 0.1 from 60 to 120 arbitrary units. We then calculated the phase portraits of AC/DC2 for each combination of maternal input concentrations (see 'Materials and methods').

The resulting phase diagram shows that, in contrast to Bcd, the position of the bifurcation boundary is entirely insensitive to Cad concentration. There is an abrupt vertical divide between dynamical regimes around 52% A–P position (*Figure 4C*). This implies that varying Cad concentration has no effect on the position of the bifurcation, and therefore the extent of the posterior region that exhibits dynamic gap domain shifts. This complements earlier studies which suggested that Cad serves as a general activator in the posterior, but is not directly involved in setting the position of gap gene expression features (*Struhl et al., 1992*; *Manu et al., 2009a*; *Verd et al., 2017*), or in controlling the rate of gap domain shifts in this region of the embryo (*Verd et al., 2018*).

## Inter-specific lability of the bifurcation boundary

The analysis described in the previous section predicts that increased levels of Bcd should lead to a posterior displacement of the bifurcation boundary between the static anterior and the shifting posterior patterning regime (*Figure 4B*). Surprisingly, exactly the opposite is observed in *M. abdita*. In this species, the anterior localization domain of *bcd* mRNA is expanded compared to *D. melanogaster*, which presumably leads to an expanded Bcd protein gradient (*Stauber et al., 1999*; *Stauber et al., 2000*; *Wotton et al., 2015*; *Crombach et al., 2016*). However, despite more Bcd being present, the bifurcation boundary is located further *anterior* than in *D. melanogaster*

(*Figure 5A*) (*Wotton et al., 2015*; *Crombach et al., 2016*). This is impossible to explain if we only take maternal inputs to the gap gene system into account.

Previous studies have shown that the strength of gap-gap cross-regulatory interactions differs between *M. abdita* and *D. melanogaster* (*Wotton et al., 2015*; *Crombach et al., 2016*). Based on this, we sought to identify the specific gap-gap interactions that could account for the altered position of the bifurcation in our models. The dissection of the gap gene network into dynamical modules narrows this search to interactions within the AC/DC1 and AC/DC2 subcircuits (*Figure 2B, C*), since their regions of influence cover the relevant region of the embryo. First, we focus on AC/DC2, because the bifurcation boundary in *D. melanogaster* lies within its region of influence. We know from previous analyses, that the strong positive feedback between *hb* and *kni* is heavily conserved between the two species (*Wotton et al., 2015*; *Crombach et al., 2016*). We also know that asymmetric interactions between overlapping gap genes (*kni* on *Kr*, and *Kr* on *hb*) are involved in regulating domain shifts in both species (*Jaeger et al., 2004b*; *Jaeger et al., 2004a*; *Ashyraliyev et al., 2009*; *Crombach et al., 2012*; *Crombach et al., 2016*). Therefore, we concentrate our analysis on these two interactions.

In our *D. melanogaster* model, repression of *hb* by Kr is negligible (*Figure 2C*). In contrast, both genetic evidence (*Wotton et al., 2015*) and models (*Crombach et al., 2016*) for *M. abdita* indicate that there is a significant net repressive effect of Kr on *hb* in this species. In *Figure 5B*, we plot the dynamical regimes of AC/DC2 in nuclei between 47 and 61% A–P position, while varying this regulatory parameter by decreasing steps of 0.0005 across a narrow range of values from to −0.0035. If repression remains minimal (closer to zero than −0.0005), we recover the bifurcation at 52% A–P position, as nuclei anterior to this position are multistable, while more posterior nuclei are monostable (*Figure 5B*). If the strength of repression is further increased, the bifurcation boundary moves anteriorly and vanishes altogether around a repression strength of −0.0025. In contrast, the position of the bifurcation boundary is largely insensitive to the interaction between *kni* and *Kr* (*Figure 5*, D). Taken together, the analysis predicts that AC/DC2 in *M. abdita* should be structurally stable, as net repression of *hb* by Kr is increased in this species (*Wotton et al., 2015*; *Crombach et al., 2016*), eliminating the bifurcation boundary present in *D. melanogaster*.

However, the bifurcation boundary in *M. abdita* is located even further anterior, as shifting boundaries extend to around 40% A–P position (*Wotton et al., 2015*; *Crombach et al., 2016*), far into the region between 35 and 45% A–P position covered by AC/DC1. Since the repression of *hb* by Kr is shared between AC/DC1 and AC/DC2, we asked if increasing its strength would induce a bifurcation in AC/DC1, abolishing its structural stability and rendering it critical. Interestingly, this is not the case, as AC/DC1 remains multistable regardless of repression strength due to the very strong positive feedback between *Kr* and *gt* (*Figure 5*). In this subcircuit, further alterations to regulatory interactions are required to render the circuit monostable. This can be achieved, for example, by simultaneously changing both repressive interactions of Kr on *hb* and Hb on *gt* (*Figure 5*). In this case, the latter interaction is the critical one. Therefore, at least two repressive interactions must be stronger in *M. abdita* than in *D. melanogaster* to render AC/DC1 critical in this species.

In summary, our analysis suggests that in *D. melanogaster* only AC/DC2 is critical, while in *M. abdita* the bifurcation boundary between static and dynamic patterning regimes falls into the region of influence of AC/DC1. This change in the stability of dynamical modules is caused by changes in the strength of particular gap-gap cross-repressive interactions. Dynamical modules and their criticality—how close they are to a bifurcation boundary—have important consequences for the evolvability of the gap gene network. Our work predicts that small changes in the strength of gap-gap cross-regulatory interactions specifically affect the extent of static versus shifting patterning regimes. In contrast, other aspects of gap gene patterning, such as the alternating-cushions mechanism of positive feedback between *Kr* and *gt* as well as *hb* and *kni*, are extremely robust towards changes in interaction strengths.

## Conclusions

Stuart Kauffman's notion of adaptive systems at the 'edge of chaos' first encapsulated the idea that evolving regulatory networks exhibit modular dynamics and are in a state of criticality (*Kauffman, 1993*). (*Irons and Monk, 2007*) later made the idea of dynamical modules precise, and formulated an algorithm to detect them in Boolean network models. Here, in the first part of our analysis, we generalize the notion of a dynamical module to continuous systems and use a pragmatic

approach based on sensitivity analysis to identify dynamical modules in the empirically tractable gap gene system of *D. melanogaster*. The three modules described include distinct sets of regulators, but share a common regulatory network structure. They all correspond to AC/DC subcircuits, able to drive multistable (switch-like) and oscillatory dynamics depending on parameter values and boundary conditions, amongst others (*Panovska-Griffiths et al., 2013*). In this paper, we show that each circuit is active in a particular region of influence along the antero-posterior axis, where it is able to reproduce the geometry of transient gap gene expression trajectories, and hence the overall expression dynamics, of a full gap gene circuit (*Figures 2* and *3*).

(*Benítez and Alvarez-Buylla, 2010*; *Benítez and Alvarez-Buylla, 2010*) have used a similar approach to study the robustness of pattern formation in the root and leaf epidermis of the mustard cress *Arabidopsis thaliana* (see also *Benítez et al., 2008*; *Benítez et al., 2011*). These authors define 'dynamic modules' in a manner quite similar to *Irons and Monk (2007)*. Their modules are composed of overlapping sets of components and interactions that generate a specific dynamic behaviour, or set of attractors, in the context of a given tissue (leaf or root). In contrast to our analysis, all of these modules are able to faithfully reproduce the behaviour of the full model. They are therefore heavily redundant and their linkage is responsible for making the patterning outcome of the whole network robust, rather than generating a diversified pattern in different regions of a tissue. This indicates that dynamical modularity can not only be used to analyse the multifunctionality of gene regulatory networks, but also the robustness or stability of their behaviour.

Dynamical modules as defined and used here should not be confused with dynamic patterning modules (*Newman and Bhat, 2008*; *Newman and Bhat, 2009*; *Hernández-Hernández et al., 2012*), which require an interaction of conserved molecular regulatory networks with generic biophysical properties of aggregated cells. Dynamic patterning modules can be considered a specific sub-type of dynamical modules, which exhibit characteristic and robust behaviour based on the self-organising reciprocal interaction of genetic and tissue-level regulation. While dynamic patterning modules are predominantly used to explain the emergence of biological form during the early evolution of multicellular organisms, dynamical modules are intended for the decomposition and functional characterization of general cellular or developmental processes and their evolutionary potential (see below).

Our dynamical modules should also not be confused with the recent postulation of 'static' and 'dynamic' modules in the segment determination network of the flour beetle *Tribolium castaneum* (*Zhu et al., 2017*). 'Static' and 'dynamic' in that case refer to slow versus rapid time scales of expression dynamics driven by distinct classes of subcircuits, which are all defined in terms of their disjoint structural components. In contrast, our results indicate that structural modularity is not essential for the evolution of insect segmentation. The search for structural modules may be in vain, as it is in the case of the gap gene network of *D. melanogaster* where only dynamical modules are present in the system. In this sense, dynamical modules provide a powerful complementary alternative to identifying structural modules.

Concerning the dynamical analysis of the AC/DC subcircuits, it is interesting to note that the geometry of transient trajectories generated by different time-variant or non-autonomous models can be equivalent despite underlying discrepancies in features of phase space (see 'Results and discussion'). In particular, posterior subcircuits AC/DC2 and AC/DC3 exhibit spiralling trajectories in the absence of spiral sink attractors (*Figure 3*, and *Figure 3B, B', C, C'*) (*Verd et al., 2017*; *Verd et al., 2018*). In these models, the spiral geometry of the trajectories is generated by a corresponding movement of a point attractor in space, while in the full model it is the consequence of (much less pronounced) attractor movement in addition to the complex eigenvalues of the spiral sink. In other words, there is a certain degeneracy or disconnect between the attractors present in phase space and the resulting transient dynamics driven by the system.

This in turn has important implications for the way in which we analyse dynamical systems models in biology. To understand the dynamics of a regulatory network, it is not generally sufficient to perform a steady-state analysis of a time-invariant or autonomous version of the system. Transient dynamics, and the explicit dependence of regulatory structure on time implied by time-variance or non-autonomy, should be considered the default assumption. Steady-state dynamics and time-invariant autonomy must first be established before conclusions from classical attractor analysis can be considered valid and applicable.

In the second part of our analysis, we focus on the consequences of dynamical modularity and criticality on the evolvability of the system. Our results shed light on the apparent paradox that gap gene expression dynamics are surprisingly insensitive to maternal gradient concentrations during development within a species, but quite labile when comparing different species on an evolutionary time scale (*Figures 4* and *5*) (*Briscoe and Small, 2015*). We show that the paradox is resolved if we consider the fact that this lability depends on changes in downstream regulatory interactions between gap genes rather than evolutionary changes to maternal gradients.

Bifurcation analysis reveals that AC/DC1 and AC/DC3 are structurally stable in *D. melanogaster*, and therefore insensitive to changes in the parameter values that affect the strength of interactions. In contrast, AC/DC2 is in a state of criticality, close to a bifurcation boundary, with regard to variation in the strength of the repressive effect of Kr on *hb*. Increasing repression between these two genes leads to a bifurcation event, which changes the dynamical regime of AC/DC2 from multistable, switch-like, behaviour (generating stable gap domain boundaries to spiralling trajectories (generating shifting transient boundaries). Evolvability of the gap gene system therefore depends on the fact that some dynamical modules are critical, while others are structurally stable. Evolution induces changes in the stability of specific modules. This explains why the extent and dynamics of gap domain shifts appear to be highly evolvable, while the basic staggered arrangement of domain boundaries remains remarkably stable, at least during the evolution of cyclorrhaphan (or 'higher') flies (*Bonneton et al., 1997*; *Stauber et al., 2000*; *Shaw et al., 2002*; *Lemke et al., 2008*; *Lemke and Schmidt-Ott, 2009*; *Lemke et al., 2010*; *Wotton et al., 2015*; *Crombach et al., 2016*, see *Jaeger, 2011* for review).

Our analysis of *D. melanogaster* allows us to infer a number of characteristics of gap gene regulation in other dipteran species, such as *M. abdita*, even though the quality of the models we have for that species unfortunately does not allow a direct comparison of phase spaces (*Crombach et al., 2012*; *Crombach et al., 2016*). Dynamic boundary shifts extend much further anterior in this species than in *D. melanogaster* (*Figure 5A*) (*Wotton et al., 2015*). The anteriorly displaced position of the bifurcation suggests that AC/DC1, not AC/DC2, must be critical in *M. abdita*. Our analysis reveals that changes in several gap-gap interactions are required to render AC/DC1 structurally unstable (*Figure 5*). This provides a plausible lineage explanation (*Calcott, 2009*) (or evolutionary trajectory) for changes in dynamical modules that accurately fit what we know about changes in gap gene expression and regulation between *M. abdita* and *D. melanogaster* (*Wotton et al., 2015*; *Crombach et al., 2016*).

Outside the cyclorrhaphan lineage, the arrangement of gap domains and the patterning output of the system changes. In the moth midge *C. albipunctata*, for example, there is no posterior domain of Gt protein expression, and the posterior domain of *hb* mRNA only forms after gastrulation (*Rohr et al., 1999*; *García-Solache et al., 2010*). Gap domain shifts are much more pronounced in this species compared to *D. melanogaster* or *M. abdita* (*García-Solache et al., 2010*; *Crombach et al., 2014*). The beetle *T. castaneum* shows an even more dynamic mode of segment determination: only the anterior-most domains form simultaneously, while more posterior domains are generated sequentially by sustained oscillations in segmentation gene expression (*Sarrazin et al., 2012*; *El-Sherif et al., 2012*).

An extended analysis of our AC/DC circuits (see Appendix 1) reveals that they can be induced to drive sustained limit-cycle oscillations with relatively small additional changes in the values of parameters that determine cross-regulatory interactions (summarized in *Figure 6*) (*Panovska-Griffiths et al., 2013*; *Perez-Carrasco et al., 2018*; *Page and Perez-Carrasco, 2018*). Although gap genes do not seem to be directly involved in the process of segment determination in *T. castaneum*, they do show repeated waves of kinematically shifting gene expression in the blastoderm and germband of the embryo (*Zhu et al., 2017*). So do pair-rule genes (*Sarrazin et al., 2012*; *El-Sherif et al., 2012*; *El-Sherif et al., 2014*), which are known to be essential for segment determination in all arthropods. Pair-rule genes in *T. castaneum* may also be regulated by AC/DC-like regulatory subcircuits (*Choe et al., 2006*). These surprising resemblances suggest that ancestral gap and pair-rule expression may have relied on similar regulatory principles, and that the regulatory changes required to turn sequential (short-germband) into simultaneous (long-germband) segmentation may be much more subtle than commonly thought (see *Tautz, 2004*; *Clark, 2017*). Improved empirical and modelling evidence from many additional species will be required to rigorously test this prediction.

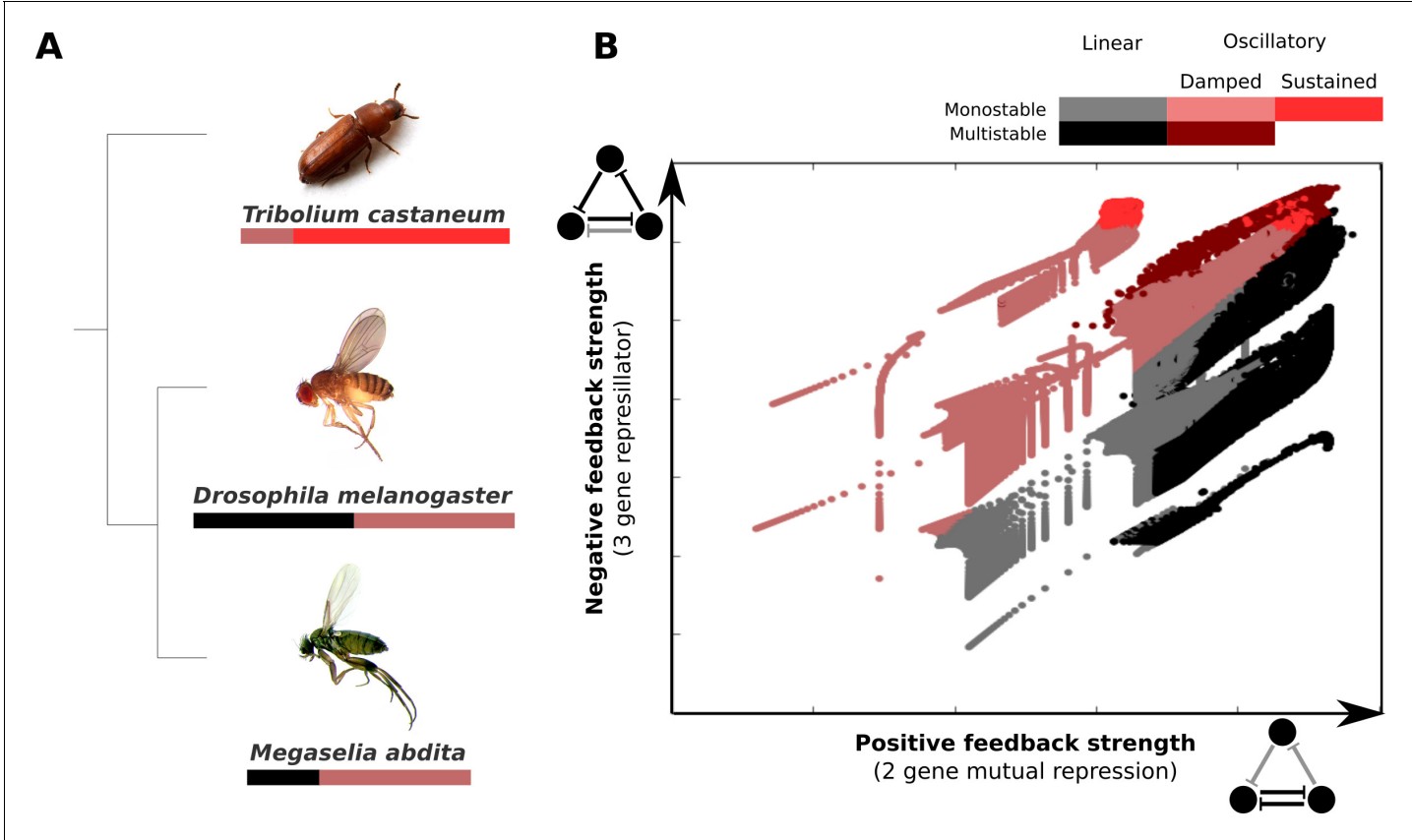

**Figure 6.** AC/DC subcircuits and their possible role in the evolution of long-germband from short-germband segment determination. (**A**) Phylogenetic relationships between the intermediate-germband insect *T. castaneum* and the two long-germband dipteran species analysed in this paper. Coloured bars indicate which dynamical regimes are active in which relative region along the A–P axis of the embryo (see key). (**B**) Visualisation of the dynamical regimes of an AC/DC circuit. We combine equation parameters into two composite control parameters which correspond to the $x$ and $\tilde{\Gamma}^3$ parameters in *Equations (28) and (29)* of Appendix 1 and represent the strength of the positive feedback between two of the genes in the circuit (X-axis) and the strength of the negative feedback involving all three genes (Y-axis) respectively. We used simplified connectionist models (see 'Materials and methods')—with all degradation rates set to equal values, time-constant basal activation terms, and no auto-activation—to evaluate the dynamical regimes for varying values of these two control parameters. The results are colour-coded as indicated in the key. Possible dynamical regimes are: monostability and multistability without (grey), and with damped oscillations (light/dark red), as well as sustained limit-cycle oscillations (bright red) (see Appendix 1 for details).

DOI: https://doi.org/10.7554/eLife.42832.009

Taken together, our results demonstrate the potential of analysing the developmental and evolutionary dynamics of regulatory networks using an approach based on dynamical rather than structural modularity. The main limitation of this approach remains the small number of systems in which this type of analysis is currently possible. While structural analysis only requires the mechanistic decomposition of a system into its components and their interactions, our analysis relies on the 'recomposition' or reconstitution of the system with a dynamical model of some sort (*Bechtel and Abrahamsen, 2005*; *Bechtel and Abrahamsen, 2010*; *Bechtel, 2011*). This model can be discrete and qualitative as in the Boolean analysis carried out by *Irons and Monk (2007)*, or it can be continuous and quantitative as in our example of the gap gene system. In the former case, we require good qualitative evidence on the expression of the relevant factors and their interactions; in the latter, we need quantitative sets of expression data suitable for model fitting (*Jaeger and Crombach, 2012*). Significant advances in 'omics' and 'big-data' techniques, genetic perturbation by gene knock-down, precise genome editing, as well as the methodologies of quantitative microscopy and image bioinformatics should render the acquisition of such evidence feasible in a wide range of cellular and developmental systems. The most important consideration for these efforts is that empirical

and modelling efforts are tightly integrated and adjusted to each other for an accurate representation and analysis of systems dynamics.

One peculiarity of our gap gene example is the fact that we are dealing with a spatio-temporal patterning system. Accordingly, the pragmatic approach for identifying dynamical modules we present here relies on the fact that different subcircuits are relevant in different regions of an embryo or tissue (see *Figure 2*). Clearly, this approach is not appropriate for regulatory systems without spatially differentiated dynamics, and in these cases a different strategy is needed. There are two possibilities worth pursuing in the regard. Analogous to the spatial case, node sensitivity analysis can be performed to identify network nodes that are not required during certain periods of time. This subdivides the system into modules active during different periods of influence, rather than regions of influence. However, this approach is limited since it will miss modules that simultaneously drive different dynamical regimes in a multistable network. To overcome this limitation in Boolean systems (whether spatial or not), the approach by *Irons and Monk (2007)* can be straightforwardly applied. It is based on mapping system components to simple and fundamental dynamical patterns (called subsystems) that combinatorially compose the attractors of a system (see also *Jaeger and Monk, 2019*). This general strategy could also be used as a guide to identifying dynamical modules in continuous systems: what are the qualitative distinguishable behaviours of the system? What are the simplest dynamical patterns they share? Which network components contribute to which of these patterns? A general strategy of this kind fits well with the type of approach we employed here in the spatial context of the gap gene system.

It is important to note that, in continuous systems, dynamical modularity will always be a matter of degree, just as in the case of structural and functional modules (*Wagner and Altenberg, 1996*). Subsystems interact and dynamical regimes depend on each other to varying degrees (see *Koseska and Bastiaens, 2017*), for an interesting perspective on the importance of this phenomenon). This is no major impediment to our approach. As long as the behaviour of a subsystem can be distinguished from other types of dynamics, we can map it to those network components and their interactions that are mainly responsible for driving it. Our analysis of the gap gene network nicely illustrates this point, as the individual AC/DC subcircuits reproduce the main features of the full system's dynamics, while subtle differences, caused by interactions between subsystems, remain detectable (see Results and Discussion).

Finally, we must ask ourselves whether our approach is scalabale to larger networks and, if so, to what extent. Scalability in this case crucially depends on the availablity of a dynamical model for the full system. If there is a Boolean model, the approach by *Irons and Monk (2007)* can be used without specific size limitations (see, for example, the networks described in *Kauffman, 1993*). For larger-scale continuous models, for instance genome-scale dynamic models of metabolism (*Smallbone and Mendes, 2013*; *Srinivasan et al., 2015*), what is needed is the ability to identify all (or at least the most prominent or biologically relevant) attractors and dynamical regimes. The numerical analysis of such models can be challenging but, given enough computing power, there is no reason to assume that it cannot be scaled to such larger network models. Indeed, since our approach depends on attractors and dynamical regimes, it is important to note that these increase in number only slowly with an increase in the number of nodes in an interaction network (*Kauffman, 1993*).

Luckily, all of the limitations discussed above are of a purely practical nature and the number of potential systems amenable to dynamical analysis is increasing. AC/DC circuits involved in neural tube patterning in vertebrates can be considered dynamical modules (*Balaskas et al., 2012*; *Panovska-Griffiths et al., 2013*; *Perez-Carrasco et al., 2018*). So can the growing number of experimentally verified Turing-type pattern generators, for example, those driving digit patterning in the vertebrate limb from sharks to mammals (*Raspopovic et al., 2014*; *Onimaru et al., 2016*). We provide a generalised account of dynamical modularity with additional examples in *Jaeger and Monk (2019)*. Apart from these examples, there are good theoretical reasons to believe that dynamical modularity is a very widespread phenomenon. The in silico screen performed by *Jiménez et al. (2017)* suggests that many regulatory systems are capable of multistable behaviours, while different dynamical regimes rarely map to cleanly separable clusters in the structure of the network. In other words, most regulatory networks are of emergent rather than hybrid type, exhibiting dynamical but not necessarily structural modularity. Based on this, we conclude that the gap gene system is probably not an isolated example of a system where our approach is useful. As a matter of fact, we see an

urgent and growing need to identify and characterise dynamical modules in development and evolution (*Jaeger and Monk, 2019*). The theory and the approach we present here greatly extend the reach of traditional methods by capturing modularity in systems that show no overt clusters in regulatory structure or co-expression patterns. They bring us closer to the aim of identifying true functional modules in evolving developmental systems (*Wagner, 2014*), as dynamical behaviour is much more tightly integrated with biological function than the regulatory structure of a network.

## Materials and methods

### The full model: a diffusion-less gap gene circuit

What we refer to as the 'full model' in this paper corresponds to a diffusion-less gap gene circuit, which is formulated in the connectionist modelling framework first proposed by *Mjolsness et al. (1991)*. It is derived from gap gene circuits with diffusion (*Ashyraliyev et al., 2009*), and was previously published and described (*Verd et al., 2017*; *Verd et al., 2018*). Here, we only provide a brief description of the model. See these previous publications for details.

Gap gene circuits consist of a one-dimensional row of nuclei, arranged along the antero-posterior (A–P axis of the embryo. They are hybrid models that implement continuous dynamics during interphase and mitosis, with discrete instantaneous nuclear divisions occurring at the end of each mitosis. The spatial domain of the model used here extends over a range of 35 to 75% A–P position (where 0% is the anterior pole), covering the trunk region of the embryo. The full model includes cleavage cycles C13 and C14A of the blastoderm stage during early development of *D. melanogaster* (*Foe and Alberts, 1983*). C14A is further subdivided into eight time classes of equal duration (T1–T8) (*Surkova et al., 2008*). Division takes place at the end of C13.

The state variables of the model represent the concentrations of transcription-factor proteins encoded by trunk gap genes *hb*, *Kr*, *kni*, and *gt*. $g_i^a(t$ represents the concentration of protein $a$ in nucleus $i$ at time $t$. The rate of change in gap protein concentration over time is given by the following system of ordinary differential equations:

$$\frac{dg_i^a(t)}{dt} = R^a \phi(u^a) - \lambda^a g_i^a(t), \tag{1}$$

where $R^a$ is the rate of protein production, and $\lambda^a$ the rate of protein decay. $\phi$ is a sigmoid regulation-expression function which is used to represent the coarse-grained saturating kinetics of transcriptional regulation. It is defined as follows:

$$\phi(u^a) = \frac{1}{2} \left( \frac{u^a}{\sqrt{(u^a)^2 + 1}} + 1 \right), \tag{2}$$

where

$$u^a = \sum_{b \in G} W^{ba} g_i^b(t) + \sum_{m \in M} E^{ma} g_i^m(t) + h^a. \tag{3}$$

$G = \{hb, Kr, kni, gt\}$ denotes the set of trunk gap genes, and $M = \{\mathrm{Bcd}, \mathrm{Cad}\}$ the set of external inputs from protein gradients encoded by maternal coordinate genes (which are not themselves regulated by gap genes. We linearly interpolate quantified spatio-temporal protein expression data (*Surkova et al., 2008*; *Pisarev et al., 2009*; *Ashyraliyev et al., 2009*) to obtain the concentration profiles of the maternal regulators $g_i^m$.

Interconnectivity matrices $W$ and $E$, with elements $w^{ba}$ and $e^{ma}$, represent regulatory weights of interactions between gap genes, and external inputs from maternal gradients, respectively. The effect of regulator $b$ or $m$ on gap gene target $a$ is activating, if the corresponding weight is positive, repressive, if the weight is negative; there is no interaction if the weight is (near zero. $h^a$ is a threshold parameter encoding the basal activity of gap gene $a$ in the absence of any spatially or temporally specific regulators. The system of *Equations (1)* governs regulatory dynamics during interphase. $R^a$ is set to zero during mitosis.

Earlier studies have established that diffusion of gap proteins is not essential for pattern formation (*Jaeger et al., 2004b*; *Manu et al., 2009a*; *Verd et al., 2017*). Therefore, it is not included in

this version of the model. Omitting diffusion renders each nucleus independent of the others, and reduces the dimensionality of system from 164 (4 gene products in 41 nuclei) to 41 independent systems with 4 dimensions each. This makes the system amenable to phase (or state) space analysis.

## Fitting of the full model

Values for parameters $R^a$, $\lambda^a$, $W$, $E$, and $h^a$ were obtained by fitting the model to quantitative spatio-temporal gene expression data as previously described (*Reinitz and Sharp, 1995*; *Jaeger et al., 2004b*; *Ashyraliyev et al., 2009*; *Verd et al., 2017*). Briefly: we solve model *Equations (1)* numerically, and compare the resulting model output to data by calculating a weighted root-mean-square (RMS) score (*Ashyraliyev et al., 2009*). This is repeated while changing parameter values until the fit is no longer improving. The difference between model output and data is minimized using a global optimization algorithm called parallel Lam Simulated Annealing (pLSA) (*Chu et al., 1999*). Model fitting was carried out on the Mare Nostrum cluster at the Barcelona Supercomputing Centre (http://www.bsc.es). The circuit used here is identical to that published and described in *Verd et al. (2017)*; *Verd et al. (2018)*. It accurately reproduces the spatio-temporal expression dynamics of gap gene expression (see *Figure 1B*), and is fully consistent with the available experimental evidence on gap gene regulation (see *Jaeger et al., 2004b*; *Jaeger et al., 2004a*; *Manu et al., 2009b*; *Manu et al., 2009a*; *Ashyraliyev et al., 2009*; *Jaeger, 2011*; *Crombach et al., 2012*; *Verd et al., 2017*; *Verd et al., 2018*).

## Identifying dynamical modules

The full gap gene circuit described above drives two distinct dynamical regimes anterior and posterior of a bifurcation, which occurs at 52% A–P position (*Verd et al., 2017*; *Verd et al., 2018*). Anterior to the bifurcation, the system is multistable, positioning static gap domain boundaries through switch-like dynamic behaviour; posterior to the bifurcation, the system is monostable, and the only attractor present is a spiral sink implementing a damped oscillator mechanism driving the observed dynamic anterior shifts of posterior gap domains (*Verd et al., 2017*; *Verd et al., 2018*). In the absence of any evident structural modules in the network (see the 'Introduction'), we ask whether there are dynamical modules or subcircuits that suffice to reproduce the observed dynamical regimes in different regions of the embryo. We define dynamical modules as subcircuits embedded in the gap gene network that are capable of recovering the dynamics of gap gene expression at a given A–P position (*Irons and Monk, 2007*). Dynamical modules may show overlap in their components, interactions, and regions of influence.

## Node sensitivity analysis

We observe that any specific nucleus along the A–P axis of the embryo only ever expresses two gap genes at the same time, and never more than three different gap genes over the whole duration of cleavage cycle C14A. This implies that only a subset of the four trunk gap genes are required for patterning at any given position. To identify these subsets and their regions of influence, we assessed the sensitivity of simulated developmental trajectories to the removal of gap genes in all nuclei between 35 and 75% A–P position. Early gap gene patterning is largely governed by regulatory inputs from maternal gradients, while gap-gap interactions become increasingly predominant during late C13 and C14A. Since our analysis focuses on gap-gap cross-regulation, we limit our analysis to C14A.

Using maternal gradient and simulated gap gene concentrations at the onset of C14A (time class T1 as initial conditions, we numerically solve the full model, and compare it to simulations where one of the gap genes and all its regulatory interactions have been erased from the model. The 'node sensitivity' of the system in a given nucleus with respect to a particular deleted gap gene (or node of the network) is then given by the distance between the trajectory simulated using the full model (curve $a$), and the trajectory obtained from a simulation lacking the node representing that gene (curve $b$). Differences are summed over all time classes (T1–T8) in C14A, which results in the following distance metric:

$$d = \frac{\sqrt{\sum_{i=1}^{8}(\mathrm{Hb}_{T_i}^a - \mathrm{Hb}_{T_i}^b)^2 + (\mathrm{Kr}_{T_i}^a - \mathrm{Kr}_{T_i}^b)^2 + (\mathrm{Kni}_{T_i}^a - \mathrm{Kni}_{T_i}^b)^2 + (\mathrm{Gt}_{T_i}^a - \mathrm{Gt}_{T_i}^b)^2}}{8}. \tag{4}$$

$(\mathrm{Hb}^j_{T_i}, \mathrm{Kr}^j_{T_i}, \mathrm{Kni}^j_{T_i}, \mathrm{Gt}^j_{T_i}$ represent simulated concentrations of gap genes at time class $T_i$ indicates whether the concentration is derived from the full model (*a*) or simulations with specific nodes removed (*b*). When a particular node is removed the concentration of that gene in the trajectory simulated from the resulting AC/DC is . The smaller the distance between trajectories, the smaller the contribution of the removed gap gene to the overall dynamics of gene expression. Sensitivities close to zero indicate that the removed gene is not required for patterning in a given nucleus during C14A. The resulting regions of insensitivity are shown in *Figure 2A*. The corresponding AC/DC subcircuits associated which each of these regions are shown in *Figure 2B and C*.

## AC/DC subcircuits

### Formulation and simulation

We implemented models of the three subcircuits consisting of the components and interactions shown in *Figure 2C* using the gene circuit modelling formalism described above. Parameter values, including time-variable maternal inputs and autoregulatory weights are taken from the full model. AC/DC1 includes *hb*, *Kr*, and *gt*, spanning the region of 35 to 47% A–P position; AC/DC2 includes *hb*, *Kr*, and *kni*, spanning the region of 49 to 59% A–P position; AC/DC3 includes *Kr*, *kni*, and *gt*, spanning the region of 61 to 75% A–P position. There is a fourth AC/DC subcircuit, consisting of *hb*, *kni*, and *gt* (AC/DC4, whose region of influence lies further posterior, outside the spatial domain of the full model. The AC/DC4 subcircuit was not analysed further in this study.

We use maternal gradients and simulated gap protein concentrations from the full model at C14-T1 as initial conditions. Simulations of AC/DC subcircuits include C14A only (no mitosis or division. During C14A, zygotic gap gene cross-regulation has become essential for the positioning of domain boundaries, largely supplanting initial positional cues by gradients of maternal activators (*Jaeger et al., 2007*; *Jaeger, 2011*).

### Phase space analysis

Since they implement time-variable maternal inputs, AC/DC subcircuits are time-variant or non-autonomous dynamical models (*Strogatz, 2015*). We previously developed a methodology to characterise and classify simulated transient trajectories in such models (*Verd et al., 2014*), which we have used to analyse the full gap gene circuit model (*Verd et al., 2017*; *Verd et al., 2018*). Here, we redeploy this method for the comparative analysis of AC/DC subcircuits and the full model. Briefly, we calculate instantaneous phase portraits for different time points by 'freezing' parameter values (maternal inputs) to the value at each given time. Steady states were calculated using a Newton-Raphson algorithm as in *Manu et al. (2009a)*. For each nucleus within an AC/DC subcircuit's region of influence, we compare trajectories and attractor positions between the subcircuit and the full model. The results are shown in *Figure 3*, and *Figure 3—figure supplement 2*, and are discussed in further detail the 'Results and discussion' section of our paper. A more systematic analysis of dynamical regimes that can be driven by the AC/DC subcircuit is provided in Appendix 1.

## Acknowledgements

We would like to thank Anton Crombach for insightful discussions, comments on the manuscript, and for providing the fitted full model used in this analysis. Other members of the Jaeger Lab in Barcelona, as well as fellows and visitors at the KLI Klosterneuburg, provided inspiring and motivating feedback on the project. We thank Erik Clark and Ruben Pérez-Carrasco for countless discussions and for their thoughtful comments on the manuscript. We thank Ovidiu Radulescu for initial discussions about the dynamical behaviour of the full circuit. The authors thankfully acknowledge the computer resources, technical expertise and assistance provided by the Barcelona Supercomputing Center—Centro Nacional de Supercomputación.

## Additional information

### Funding

| Funder | Grant reference number | Author |
|---|---|---|
| MINECO | BFU2009-10184/BFU2012-33775/SEV-2012-0208 | Johannes Jaeger |
| European Commission | FP7-KBBE-2011-5/289434 (BioPreDyn) | Johannes Jaeger |
| MEC-EMBL | | Johannes Jaeger |
| La Caixa Savings Bank | | Berta Verd |
| KLI Klosterneuburg | | Berta Verd |
| Wissenschaftskolleg zu Berlin | | Johannes Jaeger |
| Max-Planck-Gesellschaft | | Johannes Jaeger |
| Herchel Smith Fund | | Berta Verd |

The funders had no role in study design, data collection and interpretation, or the decision to submit the work for publication.

### Author contributions

Berta Verd, Conceptualization, Resources, Software, Formal analysis, Funding acquisition, Validation, Investigation, Methodology, Writing—original draft, Writing—review and editing; Nicholas AM Monk, Formal analysis, analysis of the model and revising the article critically; Johannes Jaeger, Conceptualization, Supervision, Funding acquisition, Investigation, Writing—original draft, Project administration, Writing—review and editing

### Author ORCIDs

Berta Verd https://orcid.org/0000-0001-9835-009X
Nicholas AM Monk https://orcid.org/0000-0002-5465-4857
Johannes Jaeger https://orcid.org/0000-0002-2568-2103

### Decision letter and Author response

Decision letter https://doi.org/10.7554/eLife.42832.022
Author response https://doi.org/10.7554/eLife.42832.023

## Additional files

### Supplementary files

• Transparent reporting form
DOI: https://doi.org/10.7554/eLife.42832.010

### Data availability

Gap gene expression data used to solve and fit the full model are available as supplementary information (S1_Data.ods; https://doi.org/10.1371/journal.pbio.2003174.s009) in Verd et al. (2018, PLoS Biology). They were published previously in Ashyraliev et al. (2009, PLoS Comp Biol 5: e1000548). Optimizaton and simulation code is available freely online at: https://subversion.assembla.com/svn/flysa (https://github.com/BVerd/flysa—Revision-14/tree/flysa and https://github.com/BVerd/flysa—Revision-14/tree/flyssm).

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

# Appendix 1

DOI: https://doi.org/10.7554/eLife.42832.011

## Mathematical analysis of the AC/DC circuit

The aim of this analysis is to understand the overall dynamic repertoire of the AC/DC circuit. More specifically, we want to understand the dynamical regimes (as defined in the main text that this circuit can implement with different sets of parameters determining the strength of its repressive regulatory interactions. We start from the well-studied behaviour of the three-gene repressilator circuit (**Elowitz and Leibler, 2000**; **Müller et al., 2006**; **Garcia-Ojalvo et al., 2004**; **Strelkowa and Barahona, 2010**), and then extend the analysis to the AC/DC network (**Panovska-Griffiths et al., 2013**).

For both repressilator and AC/DC circuit, we characterize the number and type of steady states present for any given set of parameter values. From this, we establish the range of different dynamical regimes that each of these circuits can implement. This comparative approach illustrates how the addition of the backward repressive interaction that distinguishes the AC/DC from the repressilator circuit affects the dynamical repertoire of the network. This analysis reveals that the AC/DC circuit (unlike the repressilator can implement all the dynamical regimes of gene expression that have been observed during segment determination in different groups of insects.

## Analysis of the repressilator circuit

Let us first consider the three-gene repressilator circuit shown in **Appendix 1—figure 1A**. Transcription factor X represses gene $y$, whose product, transcription factor Y, represses gene $z$, whose product, transcription factor Z, in turn represses gene $x$. We represent repressive regulatory interactions by bounded, monotonically decreasing, and continuous functions $G$. For example, any sigmoid-shaped regulation-expression function conforms to these general conditions.

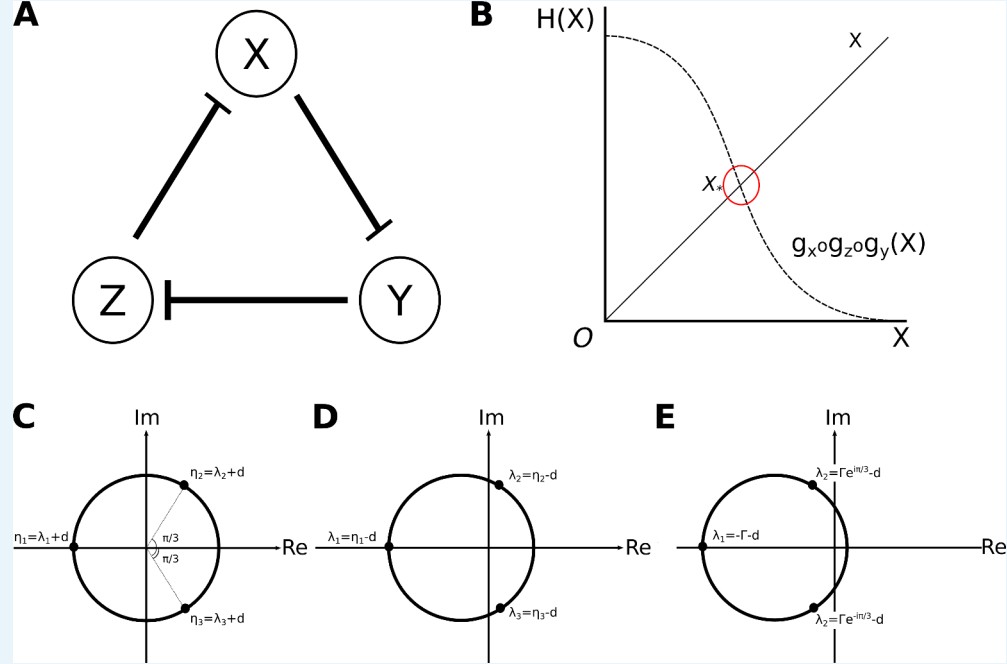

**Appendix 1—figure 1.** Analysis of the repressilator circuit. (**A**) Structure of the repressilator circuit: the nodes of the network represent genes and their products X, Y, and Z (circles).

Repressive interactions among these nodes are indicated by T-bars. We assume constitutive activation of all three nodes. (**B**) Uniqueness of solutions: solutions of **Equation (7)** are defined by intersections of the curves given by $H(X) = X$ and $H(X) = g_x \circ g_z \circ g_y(X)$. In this particular case, the shape of both curves results in them intersecting at only one value of $X$, $X_*$, and therefore there is only one solution. (**C**) Complex roots of $(\lambda + d)^3$ is a real negative number, therefore it has one real negative, and two complex roots: $\lambda_1$, $\lambda_2$ and $\lambda_3$. (**D, E**) Two different sets of possible eigenvalues: (**D**) Eigenvalues of a saddle point. One of the eigenvalues associated with the steady state has a negative real part, while the other two have positive real parts. In this case, the steady state is a saddle point. The uniqueness of the steady state, and the nature of the regulation-expression functions we've chosen, allows us to infer the presence of a limit cycle driving stable oscillations. (**E**) Eigenvalues of a stable spiral sink. All eigenvalues associated with the steady state have negative real parts and two of them are complex. In this case, the steady state is a stable spiral sink, which drives damped oscillations.

DOI: https://doi.org/10.7554/eLife.42832.012

We use the following abstract formulation of the repressilator circuit, where $G$ stands for a regulation-expression function conforming to the conditions prescribed above, and $d$ are gene product degradation rates:

$$\dot{X} = G_x(Z) - d_x X$$
$$\dot{Y} = G_y(X) - d_y Y \tag{5}$$
$$\dot{Z} = G_z(Y) - d_z Z.$$

This system has a unique steady state. Defining $g_i \equiv \frac{1}{d_i} G_i$, for $i = \{X, Y, Z\}$, the steady states of the system are solutions of the following system of equations:

$$0 = g_x(Z) - X$$
$$0 = g_y(X) - Y \tag{6}$$
$$0 = g_z(Y) - Z.$$

By solving for the state variables in the system, and substituting, we obtain:

$$X = g_x \circ g_z \circ g_y(X). \tag{7}$$

Due to the fact that the three regulation-expression functions are bounded, monotonically decreasing, and continuous, so is their composition. Therefore, there is a unique positive solution $X_*$ to **Equation (7)** (**Appendix 1—figure 1B**). By substituting backwards, following analogous reasoning, we can establish that there is a unique positive steady state given by $(X_*, Y_*, Z_*)$.

## Types of steady states: eigenvalue analysis

In order to evaluate the nature of the steady state solution of the repressilator model (**Equation 7**), we linearise the system around its steady state: $X = X_* + x$, $Y = Y_* + y$, $Z = Z_* + z$, and evaluate its eigenvalues (**Strogatz, 2015**; **Hirsch et al., 2012**). This linearisation at steady state $(X_*, Y_*, Z_*)$ is written in matrix form as:

$$\begin{bmatrix} \dot{x} \\ \dot{y} \\ \dot{z} \end{bmatrix} = \begin{bmatrix} -d_x & 0 & -\gamma_x \\ -\gamma_y & -d_y & 0 \\ 0 & -\gamma_z & -d_z \end{bmatrix} \begin{bmatrix} x \\ y \\ z \end{bmatrix}, \tag{8}$$

where

$$\gamma_x = -\frac{dG_x}{dZ}\bigg|_{Z_*}, \gamma_y = -\frac{dG_y}{dX}\bigg|_{X_*}, \gamma_z = -\frac{dG_z}{dY}\bigg|_{Y_*}. \tag{9}$$

The $\gamma_i$ are all positive due to the fact that $G_x$, $G_y$ and $G_z$ are monotonically decreasing continuous functions; they measure the sensitivity of the regulation functions at the steady

state (so, for example $\gamma_x$ measures the sensitivity of the rate of production of the product of $X$ to changes in its regulator $Z$.

The local stability of the steady state $(X_*, Y_*, Z_*$ is determined by the nature of the eigenvalues $\lambda$ associated with the linearised system at the steady state. These are the roots of the characteristic polynomial given by

$$\begin{vmatrix} -d_x - \lambda & 0 & -\gamma_x \\ -\gamma_y & -d_y - \lambda & 0 \\ 0 & -\gamma_z & -d_z - \lambda \end{vmatrix} = 0 \tag{10}$$

$$(-d_x - \lambda)(-d_y - \lambda)(-d_z - \lambda) - \gamma_x \gamma_y \gamma_z = 0 \tag{11}$$

$$(\lambda + d_x)(\lambda + d_y)(\lambda + d_z) = -\gamma_x \gamma_y \gamma_z \equiv -\Gamma^3 < 0, \tag{12}$$

where we define $\Gamma$ to be a negative real number.

For simplicity, we consider the special case where all degradation rates are equal, $d_x = d_y = d_z = d$:

$$(\lambda + d)^3 = -\Gamma^3. \tag{13}$$

The three distinct complex roots of $(\lambda + d^3 = -\Gamma^3$ are given by (see also **Appendix 1—figure 1C**):

$$\lambda + d = (-\Gamma^3)^{\frac{1}{3}} = \begin{cases} -\Gamma \\ \Gamma e^{\pm\frac{i\pi}{3}} \end{cases} \tag{14}$$

It follows that the three eigenvalues associated with the linearised system are

$$\begin{aligned} \lambda_1 &= -\Gamma - d, \\ \lambda_2 &= \Gamma e^{\frac{i\pi}{3}} - d, \\ \lambda_3 &= \Gamma e^{-\frac{i\pi}{3}} - d, \end{aligned} \tag{15}$$

$\lambda_1$ is always negative and real. However, the real parts of $\lambda_2$ and $\lambda_3$ can take either sign, since

$$Re(\lambda_{2,3}) = \frac{\Gamma}{2} - d. \tag{16}$$

Proof:

$$\begin{aligned} \lambda_2 &= \Gamma e^{\frac{i\pi}{3}} - d = \Gamma\left(\cos\left(\frac{\pi}{3}\right) + i\sin\left(\frac{\pi}{3}\right)\right) - dRe(\lambda_2) = \Gamma\cos\left(\frac{\pi}{3}\right) - d = \frac{\Gamma}{2} - d, \\ \lambda_3 &= \Gamma e^{-\frac{i\pi}{3}} - d = \Gamma\left(\cos\left(\frac{\pi}{3}\right) - i\sin\left(\frac{\pi}{3}\right)\right) - dRe(\lambda_3) = \Gamma\cos\left(\frac{\pi}{3}\right) - d = \frac{\Gamma}{2} - d. \end{aligned} \tag{17}$$

This leaves us with two alternative possibilities. From the equations in **Equation (17)** we know that $Re(\lambda_{2,3}) > 0$ if $\frac{\Gamma}{2} - d > 0$. From this, it follows that $Re(\lambda_{2,3}) > 0$ if $\Gamma > 2d$ (this case is illustrated in **Appendix 1—figure 1D**). A steady state with one negative eigenvalue and two eigenvalues with positive real parts is an unstable saddle point. Since the repressilator only has one steady state, this saddle point is unique.

By applying the Poincaré-Bendixson theorem, we can establish that the system must have a limit cycle (**Strogatz, 2015**; **Hirsch et al., 2012**). The Poincaré-Bendixson theorem in two dimensions predicts the presence of limit cycles. It tells us that if we can find a trapping region—a closed and bounded region of the phase plane that contains a trajectory that remains within this region from a certain time onwards—and this trapping region contains no equilibrium points, then there must be at least one limit cycle within this region. The Poincaré-Bendixson theorem does not normally hold for systems of dimension larger than two. However, (**Mallet-Paret and Smith, 1990**) have shown that, in monotone cyclic feedback

systems, the $\omega$-limit set of any bounded orbit can be embedded in $\mathbb{R}^2$. Under these conditions, which are met in our case, and in this two-dimensional subspace, the Poincaré-Bendixson theorem applies, and we can infer that there must be a limit cycle present. In this case, therefore, the dynamical regime of the repressilator is to drive stable limit-cycle oscillations.

There is a second possibility. If $\Gamma < 2d \Longrightarrow R_e(\lambda_{2,3})$ (**Appendix 1—figure 1E**). The fact that all three eigenvalues have negative real part, with one real and one complex conjugate pair of eigenvalues, implies that the unique steady state is a stable spiral sink. Spiral sinks are the hallmark of damped oscillations. In this case, the dynamical regime of the repressilator is to drive damped oscillations.

A third, very unlikely, possibility occurs when $Re(\lambda_{2,3} = 0$. In this case, linear stability analysis suggests that the steady state has a centre in a two dimensional subspace. However, further analysis of the full nonlinear system is required to determine the true nature of the steady state in this case.

In summary, our analysis shows that the repressilator circuit only has one unique steady state. Depending on the parameter values of the model, this steady state is either a saddle point or a spiral sink. Therefore, the dynamical repertoire of this repressilator consists of two distinct dynamical regimes: stable limit-cycle oscillations, or damped oscillations (in agreement with **Page and Perez-Carrasco, 2018**).

## Analysis of the AC/DC circuit

We obtain an AC/DC circuit (**Panovska-Griffiths et al., 2013**) by adding an additional backward repressive interaction from Z to Y to the repressilator (compare **Appendix 1—figure 1A** with **Appendix 1—figure 2A**). The mathematical formulation of the AC/DC circuit is then as follows:

$$\dot{X} = G_x(Z) - d_x X$$
$$\dot{Y} = G_y(X,Z) - d_y Y \tag{18}$$
$$\dot{Z} = G_z(Y) - d_z Z.$$

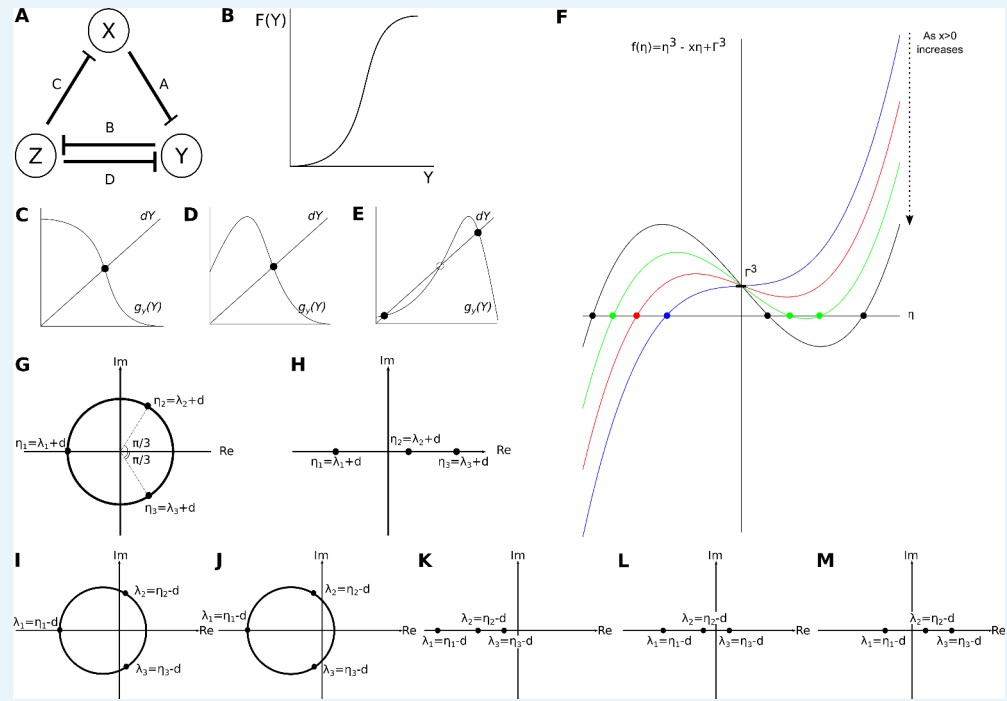

**Appendix 1—figure 2.** Analysis of the AC/DC circuit. (**A**) Structure of the AC/DC circuit: the nodes of the network represent genes and their products X, Y, and Z (circles). T-bars represent

the repressive interactions between the genes, where A,B,C and D denote the strengths of those respressions. Repressive interactions among these nodes are indicated by T-bars. We assume constitutive activation of all three nodes. (B) General shape of the function given by $F(Y)$. (C–E) Effect of increasing the back reaction on the number of steady states. Functions as defined in the text. (C) No back reaction: there is only one intersection point. (D) Weak back reaction: still, there is only one intersection point. (E) Strong back reaction: now there are three intersection points. (F) The simplified characteristic *Equation (27)* defines a depressed cubic. Its dependence on parameter values is shown. (G,H) Roots of the depressed cubic given by *Equation (27)*. (G) Option 1: one real negative root, and two complex roots with positive real part. (H) Option 2: real roots, where one has negative real part and two have positive real parts. (I–M) Combinations of eigenvalues for steady states in the AC/DC circuit. (I) Unstable spiral sink: a steady state with one real negative eigenvalue and two complex eigenvalues with positive real parts. (J) Stable spiral sink: a steady state with one real negative eigenvalue and two complex eigenvalues with negative real parts. (K) Point attractor: a steady state with real negative eigenvalues. (L) Saddle$_{1,2}$: a steady state with real eigenvalues where one is positive and two are negative. (M) Saddle$_{2,1}$: a steady state with real eigenvalues where two are positive and one is negative.

DOI: https://doi.org/10.7554/eLife.42832.013

Regulation-expression functions $G_x$ and $G_z$ are assumed to be bounded, monotonically decreasing, and continuous. $G_y(X, Z$ will typically be taken as the multiplication of the regulation-expression function of X on Y, and that of Z on Y, and its form is explained in more detail below.

Let us begin by finding the steady states of the system:

$$
\begin{aligned}
0 &= G_x(Z) - d_x X \\
0 &= G_y(X, Z) - d_y Y \\
0 &= G_z(Y) - d_z Z
\end{aligned} \Bigg\} \Rightarrow
$$

$$
\Rightarrow \begin{cases}
X_* &= \frac{1}{d_x} G_x(Z_*) = \frac{1}{d_x} G_x(\frac{1}{d_z} G_z(Y_*)) = \frac{1}{d_x} F(Y_*) \\
Y_* &= \frac{1}{d_y} G_y(X_*, Z_*) = \frac{1}{d_y} G_y(\frac{1}{d_x} F(Y_*), \frac{1}{d_z} G_z(Y_*)) \\
Z_* &= \frac{1}{d_z} G_z(Y_*)
\end{cases}
\tag{19}
$$

$Y_*$ depends on the strength of the backward reaction from Z on Y, as given by $G_z(Y_*)$, and on the combination of the reactions of Z on X and X on Y represented by $F(Y)$ (*Equation (19)*) is monotonically increasing, as it results from the composition of two monotonically decreasing functions, with a general shape as shown in *Appendix 1—figure 2B*. As the backward reaction on Y is increased, the three scenarios shown in *Appendix 1—figure 2C–E* become possible. Let us now look at the Jacobian matrix associated with the system:

$$
\begin{bmatrix}
-d_x & 0 & -\gamma_x \\
-\gamma yx & -d_y & -\gamma yz \\
0 & -\gamma_z & -d_z
\end{bmatrix}
\tag{20}
$$

where

$$
\gamma_x = -\frac{dG_x}{dZ}\bigg|_{\text{steadystate}}, \gamma_z = -\frac{dG_z}{dY}\bigg|_{\text{steadystate}} > 0,
\tag{21}
$$

and

$$
\gamma_{yx} = -\frac{dG_y}{dX}\bigg|_{\text{steadystate}}, \gamma_{yz} = -\frac{dG_y}{dZ}\bigg|_{\text{steadystate}} > 0.
\tag{22}
$$

$\gamma_{yx}$ measures the sensitivity of the repressive interaction of X on Y, and $\gamma_{yz}$, to the repressive interaction of Z on Y. Again, for the sake of simplicity, let us assume that $d_x = d_y = d_z = d$ and calculate the associated characteristic equation:

$$\begin{vmatrix} -d-\lambda & 0 & -\gamma_x \\ -\gamma_{yx} & -d-\lambda & -\gamma yz \\ 0 & -\gamma_z & -d-\lambda \end{vmatrix} = 0 \tag{23}$$

From this, it follows that

$$(-d-\lambda)^3 + (-\gamma_x)(-\gamma_{yx})(-\gamma_z) - (-d-\lambda)(-\gamma_{yz})(-\gamma_z) = 0, \tag{24}$$

$$(d+\lambda)^3 + \gamma_x \gamma_{yx} \gamma_z - (d+\lambda)\gamma_{yz}\gamma_z = 0. \tag{25}$$

If we let $d+\lambda = \eta$ and $\tilde{\Gamma}^3 = \gamma_x \gamma_{yx} \gamma_z > 0$, then **Equation (25)** becomes

$$\eta^3 - \gamma_{yz}\gamma_z \eta + \tilde{\Gamma}^3 = 0. \tag{26}$$

Now, let $x = \gamma_{yz}\gamma_z > 0$, and **Equation (26)** becomes

$$\eta^3 - x\eta + \tilde{\Gamma}^3 = 0. \tag{27}$$

This implies that the AC/DC circuit has either one or three steady states.

## Types of steady states: eigenvalue analysis

The eigenvalues of all steady states are roots of the polynomial in **Equation (27)**. In particular, $x$ and $\tilde{\Gamma}$ are evaluated at the steady state. They remain positive since

$$x = \gamma_{yz}|_{ss} \gamma_z|_{ss} > 0 \tag{28}$$

$$\tilde{\Gamma}^3 = \gamma_x|_{ss} \gamma_{yx}|_{ss} \gamma_z|_{ss} > 0. \tag{29}$$

The simplified characteristic equation shown in **Equation (27)** defines a depressed cubic (**Appendix 1—figure 2F**). Its exact shape depends on the values of $x$ and $\tilde{\Gamma}$ when evaluated at each steady state. Given that these parameters always remain positive, we can look at the shape of **Equation (27)**, and from it infer the possible roots, as well as the type and sign of every eigenvalue.

From **Appendix 1—figure 2F** we can see that, when solving for $\eta$, **Equation (27)** will always have a negative real root and, depending on the magnitude of $x$, either two positive real roots, or two conjugate complex roots, both with positive real parts (**Appendix 1—figure 2G, H**). The roots of the depressed cubic are given by $\eta_i, i \in \{1,2,3\}$, where $\eta_i = \lambda_i + d_i$ and $\lambda_i, i \in \{1,2,3\}$, are the eigenvalues associated with the corresponding steady state of the AC/DC circuit. If we consider $d_i = d, \forall i$, we get five possible combinations of eigenvalues, indicating that steady states found in the AC/DC circuit can be unstable spiral sinks (**Appendix 1—figure 2I**), stable spiral sinks (**Appendix 1—figure 2J**), point attractors (**Appendix 1—figure 2K**), saddles$_{1,2}$ (**Appendix 1—figure 2L**), or saddles$_{2,1}$ (**Appendix 1—figure 2M**).

In summary, the AC/DC circuit can exhibit dynamical regimes that consist of one or three steady states from among the five types described above. This indicates a rich dynamical repertoire, which includes the regimes shown below, known to be involved in segment determination in different groups of insects.

**Appendix 1—figures 4–6** show phase portraits that have been calculated using a connectionist version of the AC/DC model (see **Equation 33** below), where production rates $R_x = R_y = R_z = 1$. Diffusion paramters $d$, expression thresholds $h$, and repressive interaction strengths $A, B, C, D$ are shown in the corresponding **Appendix 1—tables 1–3**.

- Sustained oscillations are observed if the underlying phase portrait has a unique unstable spiral, and therefore a limit cycle (**Appendix 1—figure 3**, parameter values used for simulation are shown in **Appendix 1—table 1**).

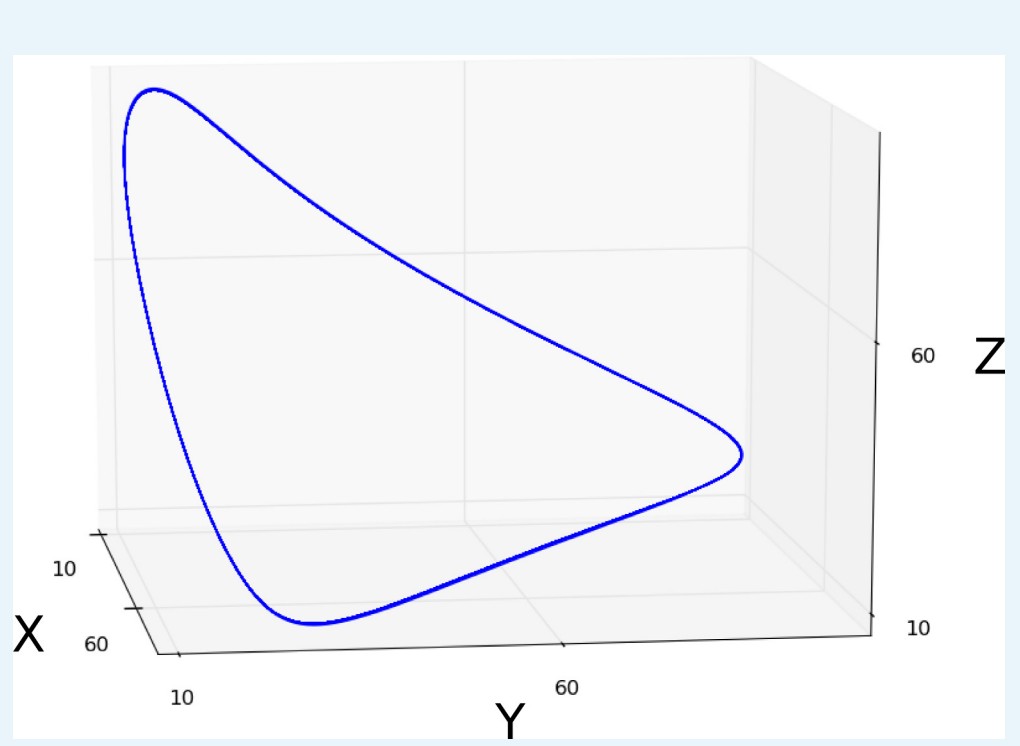

**Appendix 1—figure 3.** Sustained limit-cycle oscillations: simulation showing a limit cycle (blue curve) driving sustained oscillations in the phase portrait of an AC/DC circuit. This circuit was simulated using the connectionist formulation given by *Equation 33*, and the parameter values shown in *Appendix 1—table 1*.

DOI: https://doi.org/10.7554/eLife.42832.014

**Appendix 1—table 1.** Parameter values used to simulate the limit cycle shown in *Appendix 1—figure 3*. See *Equation 33* for parameter definitions.

| Dynamical regime | Parameters | | | | | | | |
|---|---|---|---|---|---|---|---|---|
| | $d_x$ | $d_y$ | $d_z$ | **H** | **A** | **B** | **C** | **D** |
| Sustained Oscillations | 0.09 | 0.09 | 0.09 | 1.5 | −0.01 | −0.01 | −0.01 | |

DOI: https://doi.org/10.7554/eLife.42832.015

- Damped oscillations are observed if the underlying phase portrait has a unique stable spiral sink (*Appendix 1—figure 4*, parameter values used for simulation are shown in *Appendix 1—table 2*).

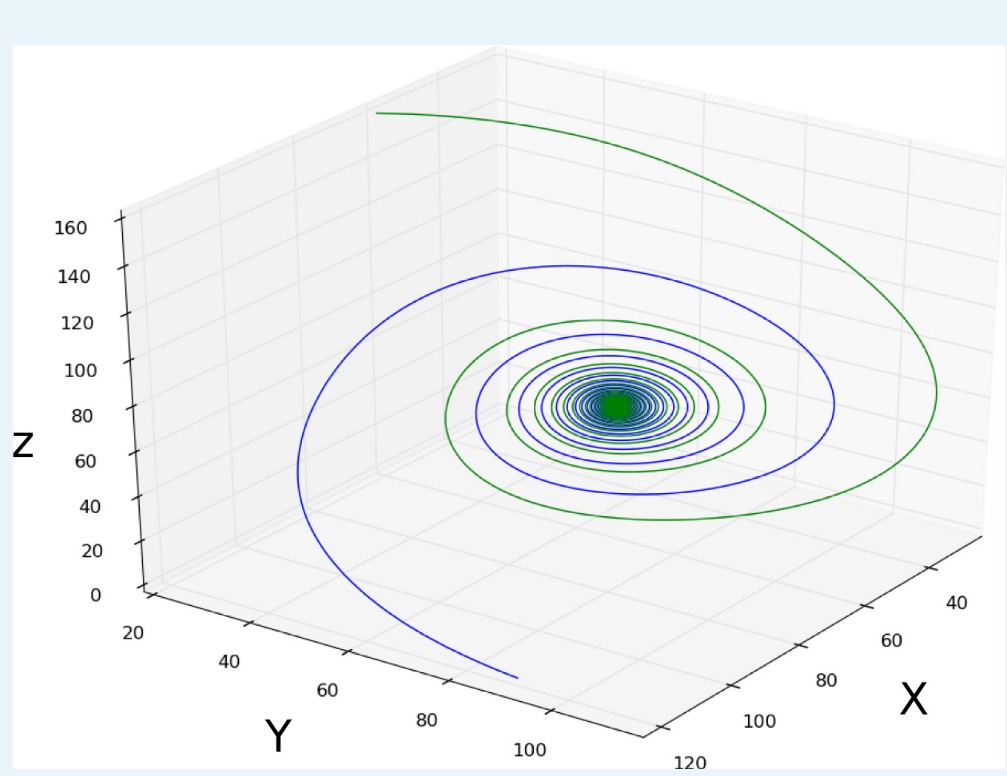

**Appendix 1—figure 4.** Damped oscillations: simulation of the damped oscillatory dynamical regime. Two spiralling trajectories (blue and green curves) are shown converging towards a stable spiral sink. This circuit was simulated using the connectionist formulation given by *Equation 33*, and the parameter values shown in *Appendix 1—table 2*.
DOI: https://doi.org/10.7554/eLife.42832.016

**Appendix 1—table 2.** Parameter values used to simulate the damped oscillations shown in *Appendix 1—figure 4*. See *Equation 33* for parameter definitions.

| Dynamical regime | Parameters | | | | | | | |
|---|---|---|---|---|---|---|---|---|
| | $d_x$ | $d_y$ | $d_z$ | **H** | **A** | **B** | **C** | **D** |
| Damped Oscillations | 0.09 | 0.09 | 0.09 | 1.5 | −0.025 | −0.025 | −0.025 | |

DOI: https://doi.org/10.7554/eLife.42832.017

- In addition, the AC/DC circuit can also exhibit a bistable dynamical regime, where the underlying phase portrait will have two point attractors and a saddle$_{1,2}$ (*Appendix 1—figure 5*, parameter values used for simulation are shown in *Appendix 1—table 3*).

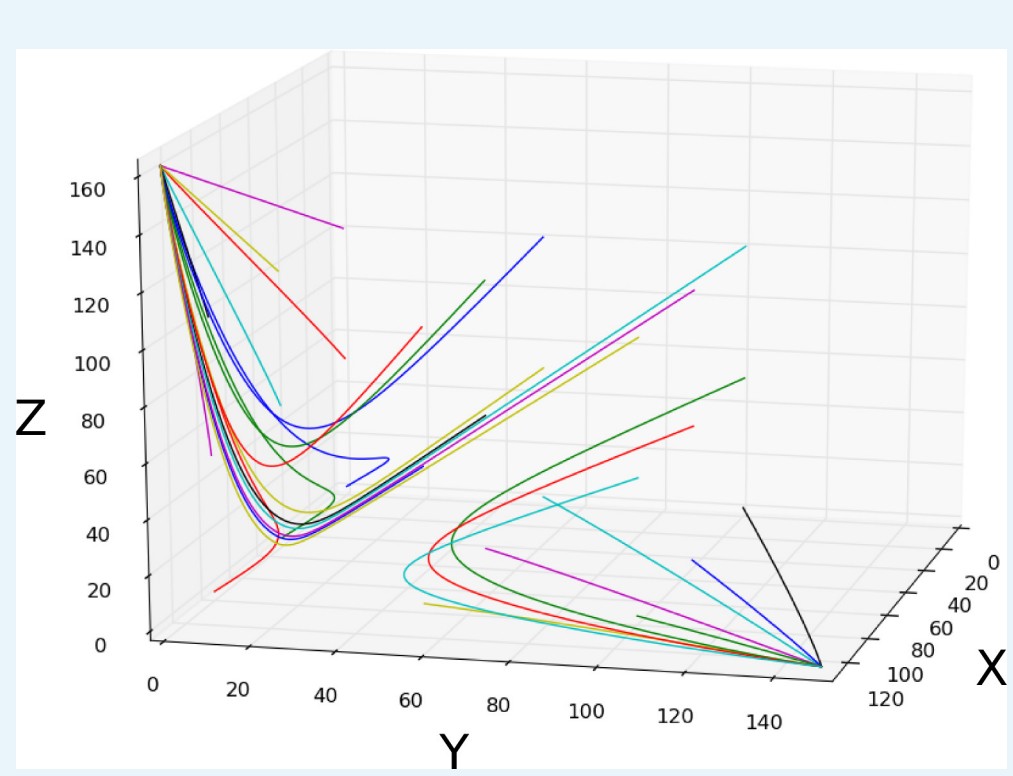

**Appendix 1—figure 5.** Bistability: simulation of the bistable dynamical regime. Trajectories with varying initial conditions (coloured curves) are shown to converge towards the two point attractors. This circuit was simulated using the connectionist formulation given by *Equation 33*, and the parameter values shown in *Appendix 1—table 3*.
DOI: https://doi.org/10.7554/eLife.42832.018

**Appendix 1—table 3.** Parameter values used to simulate the bistable regime shown in *Appendix 1—figure 4*. See *Equation 33* for parameter definitions.

| Dynamical regime | Parameters | | | | | | | |
|---|---|---|---|---|---|---|---|---|
| | $d_x$ | $d_y$ | $d_z$ | **H** | **A** | **B** | **C** | **D** |
| Bi-stability | 0.09 | 0.09 | 0.09 | 1.5 | −0.01 | −0.09 | −0.01 | −0.09 |

DOI: https://doi.org/10.7554/eLife.42832.019

We were able to find additional dynamical regimes by sampling the parameter space numerically. These are listed below. Note that a numerical analysis cannot guarantee an exhaustive list of all possible dynamical regimes. The biological relevance of these regimes has not yet been explored.

- An unstable spiral can co-exist with a saddle and a point attractor.
- A stable spiral sink can co-exist with a $saddle_{2,1}$ and a point attractor in a bistable regime. The saddle probably needs to have a 2-dimensional unstable manifold, for there to be a basin of attraction with a stable spiral.
- A bistable regime with two stable spiral sinks.

We conclude that the dynamical repertoire of the AC/DC circuit contains at least six different monostable and multistable regimes, three of which correspond to behaviour observed in insect segmentation. Similar dynamical regimes have been observed when the ACDC circuit was explored using different regulatory functions (*Perez-Carrasco et al., 2018*; *Page and Perez-Carrasco, 2018*).

## Validation of the analysis when using a connectionist model formulation

The analysis above uses an abstract general formulation of the AC/DC circuit to characterize its dynamical repertoire. We validate this analysis for the connectionist formalism as described in 'Materials and methods', which was used for all the simulations in the main paper.

Let us consider the AC/DC circuit shown in *Appendix 1—figure 6A*, and this time formulate it mathematically as follows:

$$\dot{X} = R_x g(u) - d_x X$$
$$\dot{Y} = R_y g(u) - d_y Y \quad (30)$$
$$\dot{Z} = R_z g(u) - d_z Z$$

where:

$$g(u) = 0.5\left(\frac{u}{\sqrt{u^2+1}} + 1\right) \quad (31)$$

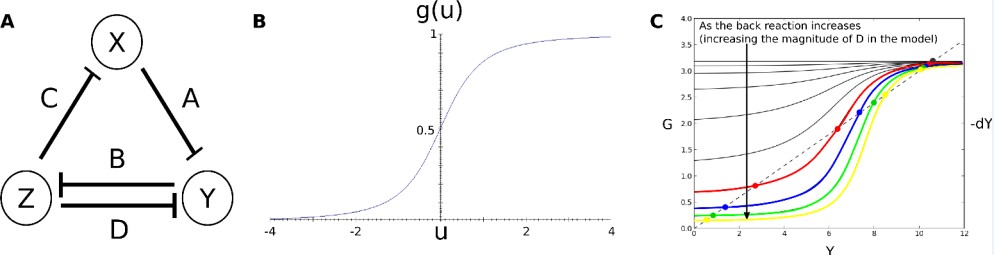

**Appendix 1—figure 6.** Connectionist AC/DC circuit. (**A**) Regulatory structure and parameters of the simulated AD/DC circuit. Nodes represent genes and their transcription-factor products. T-bars represent repressive regulatory interactions. All genes are constitutively activated. (**B**) Sigmoidal regulation-expression function used for the simulations (*Equation 31*). (**C**) The number of steady states in the connectionist AD/DC circuit, can be one (black curves), or three (coloured curves) depending on the strength of the back reaction.

DOI: https://doi.org/10.7554/eLife.42832.020

The sigmoidal function *Equation (31)* is shown in *Appendix 1—figure 6B*. It is bounded, monotonically increasing, and continuous. $u$ represents the sum of all regulatory contributions to a given target gene. For example, the contribution of gene product X on gene y is evaluated as the product between the strength of the interaction of X on y, the sign of that interaction (negative for repression, positive for activation, denoted by $\delta_{+-}$), and the concentration of X. $R$ represents the maximum rate of activation for every gene. More generally, we write $u$ as:

$$u = \sum_i (\delta_{+-})\alpha_i X_i, \quad (32)$$

based on which *Equation (30)* becomes:

$$\dot{X} = R_x g(-CZ + h) - d_x X$$
$$\dot{Y} = R_y g(-AX - DZ + h) - d_y Y \quad (33)$$
$$\dot{Z} = R_y g(-BY + h) - d_z Z$$

In what follows, we will assume that $R_x = R_y = R_z = R$, $d_x = d_y = d_z = d$, and $\frac{R}{d} = \bar{R}$. As in the general case explored above, it is possible to have more than one steady state with the connectionist formulation of an AC/DC circuit:

$$
\left.
\begin{aligned}
0 &= R_x g(-CZ + h) - d_x X \\
0 &= R_y g(-AX - DZ + h) - d_y Y \\
0 &= R_z g(-BY + h) - d_z Z
\end{aligned}
\right\} \Rightarrow
$$

$$
\Rightarrow
\begin{cases}
X_* &= \bar{R} g(-C\bar{R} g(-BY_* + h) + h)) \\
&= \bar{R} F(Y_*) \\
Y_* &= \bar{R} g(-A\bar{R} g(-C\bar{R} g(-BY_* + h) + h) - D\bar{R} g(-BY_* + h) + h) \\
&= \bar{R} G(Y_*) \\
Z_* &= \bar{R} g(-BY_* + h)
\end{cases}
\tag{34}
$$

Again, $F(Y$ is monotonically increasing ($F'(Y) \geq 0, \forall Y$), since it is the composition of two monotonically decreasing functions ($g$ is increasing with respect to $u$, but decreasing with respect to X, Y and Z). Depending on the strength of the back reaction—which is given by the magnitude of parameter $D$ in the connectionist model formulation (**Appendix 1—figure 6A**)—we will have one or three steady states (**Appendix 1—figure 6C**). This is in agreement with our general analysis above.

We will now show that the characteristic equation is the same in both cases. Once this is established, it follows that the same types of steady states occur in both model formulations. Therefore, the dynamical regimes revealed by our general analysis also apply in the connectionist case. Let us look at the associated Jacobian

$$
\begin{bmatrix}
-d_x & 0 & -\gamma_x \\
-\gamma_{yx} & -d_y & -\gamma_{yz} \\
0 & -\gamma_z & -d_z
\end{bmatrix}
\tag{35}
$$

where

$$
\begin{aligned}
\frac{\partial [R_x g(-CZ + h) - d_x X]}{\partial Z} &= \frac{dR_x g(u)}{du} \frac{du}{dZ} \\
&= \frac{dR_x g(u)}{du} \frac{d(-CZ + h)}{dZ} \\
&= \overline{\gamma_x}(-C) = -\gamma_x < 0
\end{aligned}
\tag{36}
$$

$$
\begin{aligned}
\frac{\partial [R_z g(-BY + h) - d_z Z]}{\partial Y} &= \frac{dR_z g(u)}{du} \frac{du}{dY} \\
&= \frac{dR_z g(u)}{du} \frac{d(-BY + h)}{dY} \\
&= \overline{\gamma_z}(-B) = -\gamma_z < 0
\end{aligned}
\tag{37}
$$

$$
\begin{aligned}
\frac{\partial [R_y g(-AX - DZ + h) - d_y Y]}{\partial X} &= \frac{dR_y g(u)}{du} \frac{\partial u}{\partial X} \\
&= \frac{dR_y g(u)}{du} \frac{\partial(-AX - DZ + h)}{\partial X} \\
&= \overline{\gamma_{yx}}(-A) = -\gamma_{yx} < 0
\end{aligned}
\tag{38}
$$

$$
\begin{aligned}
\frac{\partial [R_y g(-AX - DZ + h) - d_y Y]}{\partial Z} &= \frac{dR_y g(u)}{du} \frac{\partial u}{\partial Z} \\
&= \frac{dR_y g(u)}{du} \frac{\partial(-AX - DZ + h)}{\partial Z} \\
&= \overline{\gamma_{yz}}(-D) = -\gamma_{yz} < 0.
\end{aligned}
\tag{39}
$$

If we now assume that $d_x = d_y = d_z = d$, we have the same characteristic equation for the connectionist formulation as in the general case (see **Equation 23**):

$$\begin{vmatrix} -d-\lambda & 0 & -\gamma_x \\ -\gamma_{yx} & -d-\lambda & -\gamma_{yz} \\ 0 & -\gamma_z & -d-\lambda \end{vmatrix} = 0 \tag{40}$$

This demonstrates that the results of the analysis of a general AC/DC circuit are still valid when the AC/DC circuit is formulated as a connectionist model. Since the characteristic equation is the same in both cases, the dynamical repertoire will be equivalent as well.

