## [Decision Letter]

Thank you for submitting your article "Modularity, criticality, and evolvability of a developmental gene regulatory network" for consideration by *eLife*. Your article has been reviewed by three peer reviewers, who are generally positive about the paper, but raise some important issues that need to be addressed, and the evaluation has been overseen by a Reviewing Editor and Patricia Wittkopp as the Senior Editor. The reviewers have opted to remain anonymous.

The reviewers have discussed the reviews with one another and the Reviewing Editor has drafted this decision to help you prepare a revised submission.

Summary:

The three reviewers found much to like in this work and I thought the authors would appreciate hearing this in their own words, which I've combined together below. Following these summary comments is a single list of required revisions:

The manuscript "Modularity, criticality, and evolvability of a developmental gene regulatory network" is a theoretical work, based on previously published data and models. It is first concerned with the "modularity" keyword of the title: it presents a method to detect functional modules of interaction networks. The method defines dynamical modules as those where one or more of the nodes of the network are largely negligible and the whole network behaviour can be reasonable well captured by a reduced module without the irrelevant nodes.

Then it applies this method to the gap gene regulatory network of the vinegar fly *Drosophila melanogaster*, finding it can be simplified into simpler modules along the antero-posterior axis of the fly embryo and using these modules to analyse the system. This analysis draws conclusions on the other two keywords, "criticality and evolvability". It has long been discussed how different kind of networks, including gene regulatory ones, are "critical" or in a state of "criticality". This means very different things in different contexts, form properties of the topology of the networks to dynamical properties. Here a clear definition of criticality is made: criticality is to be "close to a bifurcation point beyond which" the "dynamics may change drastically and abruptly". The study shows that one of the modules of the gap gene network is critical, actually, at both sides of a bifurcation depending on the position along the antero-posterior axis of the fly embryo. This has implications on evolvability: small changes in the critical module might be the source of phenotypic change. To discuss this, *D. melanogaster* is compared to *Megaselia abdita*, showing that indeed the developmental differences between these flies can be explained in terms of the critical properties (bifurcations) of their regulatory networks.

The "modularity" part, with the definition of "dynamical modules" of a network as opposed to the "structural modules" found in the literature, is a clever trick that works very well in this systems, but I am afraid it might not be easy to generalise. The model used here is actually 41 independent networks of 4 nodes each. In this setting, it is relevant to find in which of those 41 networks we may leave out one or more nodes. However, in the more frequent modelling situation where we have a single network, this strategy would only single out nodes that are irrelevant and should have not been in the model to start with. It might be useful, however, in non-autonomous systems, to define different dynamical modules as time progresses.

However, I find the "criticality and evolvability" conclusions highly relevant. It does not depend on the modular decomposition of the network: the full model already shows the bifurcation discussed in the AC/DC2 module. However, pointing at the importance of dynamical properties as bifurcations with regard to the evolvability of developmental systems is a strong message, and it is cleanly shown in this example. I believe this way of seeing the underlying mechanics of evo-devo showcases a very generic result, relevant for the whole field, and as such worth of attention and the exposure that a publication in *eLife* would bring. For this reason I recommend publication.

This manuscript details a very important finding: subnetworks with largely overlapping genes can display functional modularity and evolvability. This conclusion complements the current views in the literature that focus on structural modularity, in which modular networks don't share genes. The results helps explain a common situation, in which genes are implicated in multiple related networks that act at different times, and hence would appear to have many pleiotropic effects. The results illustrate a compelling example of functional modularity, and I found the explanations of rather complex modeling quite clear such that a general readership could follow the logic and main points.

The paper by Verd and collaborators also offers a new angle to the study of modularity in molecular networks. Specifically, it contributes to developing the concept of dynamic modularity and the relation between phenotypes and dynamic modules. The authors were able to formally identify the repetition of a dynamic subcircuit in the gap gene network, and to study the dynamic behavior of each subcircuit in a non-autonomous context. I found particularly relevant their results and discussion on the potential mechanisms for the evolution and evolvability of the gap gene network, for which a comparative AND dynamic approach was followed. The careful analysis of dynamic systems that are partially shared by different lineages is a much needed perspective in evo-devo and is thus an important contribution of the paper.

I would like to discuss the suitability of these "technical" works for multidisciplinary journals. Nobody would comment on how "technical" an experimental effort would be, no matter how obscure its methods and how subtle and prone to misconception the results might be for a reader not absolutely familiar with the experimental technique used. However, theory papers based on mathematical ideas have to fight again and again against the prejudice of being "technical", no matter how clearly the premises and results are discussed and made amenable for a general audience. This has to come to an end if we expect the life sciences to one day be a truly multidisciplinary effort. Only the importance of the question made and the relevance of the conclusions, together with the clarity of presentation, should matter; the method used (experiment or theory) should be irrelevant as long as it can be assessed to be sound and correct. With respect to this, this manuscript is well written and makes its points clearly, and the authors have rightly considered that *eLife* was a reasonable venue to present their ideas and results.

Essential revisions:

1) My main suggestion is that the authors provide a little more guidance to back up their impact statement, which suggests that the paper provides a road map for others to apply this approach. Typical readers would not necessarily know whether they themselves can apply this approach in other systems. The authors acknowledge that few systems have enough empirical knowledge to apply these models. Is this paper meant to be a cautionary example that separate processes with overlapping genes could still act as functional modules? Or is this meant to be a roadmap to researchers? If the authors intend a roadmap, I would recommend adding to the conclusions a more explicit statement of what empirical data researchers would need in order to apply this approach. In either case, I suggest more extensive discussion of what this means for a reader who doesn't study gap genes in insects. What does non-modularity look like? Summarizing all the parameter sensitivity analyses, is it likely than any overlapping AC/DC circuits would exhibit functional modularity, or is this likely a special case? Is functional modularity a yes-no categorization, or are there degrees of functional modularity (e.g. do the departures from the full model imply some lack of functional modularity, and does that matter?). Adding a bit more to guide the reader here will clarify the broader implications of the work.

2) Another point that could benefit from more basic discussion is the link between criticality and evolvability. Criticality is likely a hard topic for the readers to picture. Perhaps one or more phase portraits could be added to a figure to help illustrate how features of the dynamical system promote evolvability? The Discussion is clear in general, but more high level summary statements could help some readers understand what dynamical systems would reduce evolvability and what would promote evolvability.

3) Another potentially challenging point is the non-autonomous aspect of the models. Perhaps a simpler statement when this first appears can help the reader understand autonomy more quickly and intuitively.

4) As the authors mention, their study model constitutes a very small network. Could they comment or speculate on how robust and/or dependent on the network scale and type of data is the dynamic vs. structural identification of modules?

5) How likely is it to follow a similar, pragmatic, approach for the identification of modules in other systems (it would be useful to point to some examples)? Is this approach not possible only because of the structural or dynamic peculiarities of the gap gene system (one of the two or three best characterized gene networks)? If such pragmatic approach were not widely applicable to other systems, it would be valuable to elaborate on what this case study can contribute to the more general dynamical/structural module study and on potential ways to test these ideas in other systems. For example, how could one test that identical transient behavior can be caused by different types of moving attractors in biological non-autonomous dynamical systems in other networks that are not as well studied.

---

## [Author Response]

Essential revisions:1) My main suggestion is that the authors provide a little more guidance to back up their impact statement, which suggests that the paper provides a road map for others to apply this approach. Typical readers would not necessarily know whether they themselves can apply this approach in other systems. The authors acknowledge that few systems have enough empirical knowledge to apply these models. Is this paper meant to be a cautionary example that separate processes with overlapping genes could still act as functional modules? Or is this meant to be a roadmap to researchers? If the authors intend a roadmap, I would recommend adding to the conclusions a more explicit statement of what empirical data researchers would need in order to apply this approach. In either case, I suggest more extensive discussion of what this means for a reader who doesn't study gap genes in insects. What does non-modularity look like? Summarizing all the parameter sensitivity analyses, is it likely than any overlapping AC/DC circuits would exhibit functional modularity, or is this likely a special case? Is functional modularity a yes-no categorization, or are there degrees of functional modularity (e.g. do the departures from the full model imply some lack of functional modularity, and does that matter?). Adding a bit more to guide the reader here will clarify the broader implications of the work.

We definitely intended our paper to be a roadmap, the beginning of a new research programme, and not a cautionary tale. We have added four new paragraphs to our “Conclusions” section of the manuscript, which discuss what type of empirical evidence is required to apply our approach, how it can be applied to systems that are not spatially differentiated, and how it scales to the analysis of larger networks (see also points 4 and 5 below).

We emphasise that non-modularity manifests itself in the absence of structural clusters in a multifunctional network. This is a very general criterion to identify systems with dynamical, but not structural, modularity.

We have also added a paragraph which discusses the fact that modularity (whether functional, structural, or dynamical) is always a matter of degree, at least in continuous dynamical systems.

We hope that these discussions clarify the broader implications and applicability of our work.

2) Another point that could benefit from more basic discussion is the link between criticality and evolvability. Criticality is likely a hard topic for the readers to picture. Perhaps one or more phase portraits could be added to a figure to help illustrate how features of the dynamical system promote evolvability? The Discussion is clear in general, but more high level summary statements could help some readers understand what dynamical systems would reduce evolvability and what would promote evolvability.

We have added several clarifying statements (Results and Discussion, and in the Conclusions) about the role of criticality and structural stability in determining the evolvability of a regulatory systems. The discussion, in particular in “Conclusions”, is intended to make the different sensitivities of subsystems understandable to the reader who is not familiar with dynamical systems theory.

3) Another potentially challenging point is the non-autonomous aspect of the models. Perhaps a simpler statement when this first appears can help the reader understand autonomy more quickly and intuitively.

We now introduce the concept of time-variance or non-autonomy more gently. Since “non-autonomy” is a potentially confusing term, we now use “time-variant” along with it throughout the manuscript. This both clarifies the concept and maintains continuity with our earlier publications on non-autonomous systems.

4) As the authors mention, their study model constitutes a very small network. Could they comment or speculate on how robust and/or dependent on the network scale and type of data is the dynamic vs. structural identification of modules?

We have added a paragraph to the “Conclusions” section discussing how our approach scales to larger systems.

5) How likely is it to follow a similar, pragmatic, approach for the identification of modules in other systems (it would be useful to point to some examples)? Is this approach not possible only because of the structural or dynamic peculiarities of the gap gene system (one of the two or three best characterized gene networks)? If such pragmatic approach were not widely applicable to other systems, it would be valuable to elaborate on what this case study can contribute to the more general dynamical/structural module study and on potential ways to test these ideas in other systems. For example, how could one test that identical transient behavior can be caused by different types of moving attractors in biological non-autonomous dynamical systems in other networks that are not as well studied.

As mentioned for point 1 above, we have added four paragraphs to the “Conclusions” section, that discuss how our approach can be applied to other systems, even if they are quite different in nature/scale to the example of the gap gene network which we use in this paper.